# Calcium bursts allow rapid reorganization of EFhD2/Swip-1 cross-linked actin networks in epithelial wound closure

Franziska Lehne [1], Thomas Pokrant[2], Sabnam Parbin[3], Gabriela Salinas[3], Jörg Großhans[4], Katja Rust[1], Jan Faix [2] & Sven Bogdan [1✉]

Changes in cell morphology require the dynamic remodeling of the actin cytoskeleton. Calcium fluxes have been suggested as an important signal to rapidly relay information to the actin cytoskeleton, but the underlying mechanisms remain poorly understood. Here, we identify the EF-hand domain containing protein EFhD2/Swip-1 as a conserved lamellipodial protein strongly upregulated in *Drosophila* macrophages at the onset of metamorphosis when macrophage behavior shifts from quiescent to migratory state. Loss- and gain-of-function analysis confirm a critical function of EFhD2/Swip-1 in lamellipodial cell migration in fly and mouse melanoma cells. Contrary to previous assumptions, TIRF-analyses unambiguously demonstrate that EFhD2/Swip-1 proteins efficiently cross-link actin filaments in a calcium-dependent manner. Using a single-cell wounding model, we show that EFhD2/Swip-1 promotes wound closure in a calcium-dependent manner. Mechanistically, our data suggest that transient calcium bursts reduce EFhD2/Swip-1 cross-linking activity and thereby promote rapid reorganization of existing actin networks to drive epithelial wound closure.

[1] Institute of Physiology and Pathophysiology, Department of Molecular Cell Physiology, Philipps-University Marburg, Marburg, Germany. [2] Institute for Biophysical Chemistry, Hannover Medical School, Hannover, Germany. [3] NGS-Integrative Genomics Core Unit, Department of Human Genetics, University Medical Center Göttingen, Göttingen, Germany. [4] Department of Biology, Philipps-University Marburg, Marburg, Germany. ✉email: sven.bogdan@staff.uni-marburg.de

The actin cytoskeleton provides the mechanical forces driving cell shape changes and cell migration[1]. Cells have evolved more than 60 families of actin-binding proteins (ABPs) that maintain the pool of actin monomers and promote actin nucleation, elongation, severing, and cross-linking of filaments[2]. At the leading edge of migrating cells (the so-called lamellipodium), a dense network of actin filaments is nucleated and organized in branched arrays by the Arp2/3 complex, which is activated by nucleation-promoting factors[3] such as WASP and WAVE proteins[3–7]. Arp2/3-mediated actin polymerization is referred to as dendritic nucleation by which branched actin networks generate actin pushing forces against the membrane surface[8]. Force generation in lamellipodia further depends on FMNL 2/3 formins and Ena/VASP family actin polymerases[7–9].

3D-electron tomography analysis further suggested an additional important role of cross-linking proteins as mechanical elements that stabilize the actin network in the lamellipodium[10–12]. In response to mechanical or biochemical cues, dynamic changes in cross-linked filaments might further allow the migrating cell to rapidly alter its mechanics by converting highly cross-linked, elastic networks into weakly cross-linked, viscous networks[13]. Different classes of actin-binding proteins such as filamin, plastins, α-actinin, or fascin present in Arp2/3-dependent lamellipodia contribute to the stabilization of lamellipodium architecture and embedded actin bundles[14–21]. These proteins frequently exhibit a dual activity by either cross-linking filaments into loose networks or laterally into tight bundles of filaments in vitro, often depending on protein concentration. In vivo, actin bundles might be preferentially induced by a short actin cross-linker, such as fascin, whereas long cross-linkers with a long spacer between the actin-binding sites, such as filamin, rather cross-link the actin networks[2,22].

Some of these actin-binding proteins additionally contain calcium-binding EF-hands[23,24]. Although the role of calcium in regulating cell shape and cell migration has been long recognized, it remained largely elusive, how EF-hand domain-containing actin-binding proteins regulate protrusion dynamics and cell motility by reorganization of the actin cytoskeleton in a $Ca^{2+}$-dependent manner[24,25]. The EF-hand domain-containing EFhD2/Swip-1 protein has been previously described as a prime candidate for controlling calcium-depending actin bundling in different tissues across species[26–32]. EFhD2/Swip-1 was found significantly upregulated in a number of pathological conditions of inflammation, immune response, and cancer suggesting an important role of EFhD2/Swip-1 in cell migration and cell adhesion during immune cell response and cancer invasion[30,32–35].

Here we identify EFhD2/Swip-1 as a calcium-dependent actin cross-linker required for lamellipodial protrusions in cell migration and epithelial wound response. Our data mechanically explain how calcium bursts allow a rapid reorganization of existing actin networks to drive epithelial wound closure.

## Results

**Swip-1 is upregulated in activated *Drosophila* macrophages.** *Drosophila* macrophages undergo a remarkable shift from a quiescent sessile state to highly polarized migratory cells at the onset of metamorphosis[36,37]. Comparative RNA-seq gene expression analysis revealed 1542 differentially regulated genes, from which 804 genes are up-regulated in highly motile pupal macrophages compared to quiescent larval macrophages (Supplementary Fig. 1a). Among the top 50 cytoskeletal/motility candidates upregulated in pupal macrophages, we identified genes encoding known pro-migratory proteins such as FHOD1/Fhos[38], different subunits of the Arp2/3 complex, and the WAVE

Regulatory Complex (WRC) as well as several actin filament-binding proteins including the EF-hand domain-containing Swip-1 protein (Supplementary Fig. 1b). We focused our attention on Swip-1, which was significantly enriched in prepupal macrophages with a fold change[28] of 2.46 (Supplementary Fig. 1b). In variance to mammalian cells, *Drosophila* only contains a single gene encoding Swip-1 (Supplementary Fig. 1c). The domain structure of *Drosophila* Swip-1 protein resembles its mammalian orthologs, EFhD1 (also termed Swip-2) and EFhD2 (also termed Swip-1) bearing two highly conserved EF-hands and a C-terminal coiled-coil domain (ref. [39] and Supplementary Fig. 1c).

**Swip-1 localizes to protruding lamellipodia of macrophages.** Ectopically expressed human EFhD2 has been found to localize to lamellipodial protrusions in motile cultured cells[30,40,41]. Consistently, we found that endogenous Swip-1 is highly abundant in protruding lamellipodia of *Drosophila* macrophages (Fig. 1a, a', c). A complete loss of immunostaining in RNAi-depleted cells confirmed the high specificity of the anti-Swip-1 antibody (Fig. 1a', b). Swip-1 displayed a prominent colocalization with F-actin in the lamellipodia (Pearson's correlation coefficient: 0.68 ± 0.05; Manders' colocalization coefficient: 85.3%). High-resolution structured illumination microscopy (SIM) analysis further revealed a striking localization of Swip-1 in a punctate pattern along actin filament bundles and cross-links (Fig. 1c, d). To visualize the dynamic localization of Swip-1 in protruding lamellipodia we generated transgenic flies expressing an eGFP-tagged protein. Spinning disk microscopy imaging of transgenic prepupae revealed a dynamic and broad localization of Swip-1-eGFP in protruding lamellipodia of crawling macrophages (Fig. 1e, e' and Supplementary Movie 1). A localization in lamellipodia was also found in macrophage-like *Drosophila* S2R +cells[42], transiently expressing eGFP-tagged Swip-1 (Fig. 1f). Interestingly, forced strong overexpression of Swip-1 even promoted lamellipodia formation as confirmed by quantification of increased lamellipodial width of transfected S2R+ cells compared to cells only transfected with eGFP control plasmids (Fig. 1g). Thus, Swip-1 localizes to protruding lamellipodia comparable to mammalian cells.

**Swip-1 promotes lamellipodia formation and controls macrophage migration.** Since overexpression of Swip-1 promoted lamellipodia formation, we next sought to analyze the consequences of loss of *swip-1* function, which has yet not been addressed in *Drosophila*. The *Drosophila swip-1* gene is located on the second chromosome corresponding to the cytological location 37B8 (Fig. 2a). It consists of two exons encoding a 25 kDa protein. We took advantage of CRISPR/Cas9-mediated genome editing to introduce small deletions within the first exon of the *swip-1* gene locus (Fig. 2a). We isolated several viable frame-shift mutants resulting in a complete loss of Swip-1 protein (Fig. 2a, b). Macrophages isolated from mutant pupae showed an impaired spread morphology with a reduced lamellipodia width and a reduced circularity compared to wild type (Fig. 2c, d; quantification in Fig. 2e, f). Re-expression of Swip-1 substantially rescued defects in the spread morphology of mutant macrophages (Fig. 2f, g).

Noticeable differences were also seen in the motility of mutant cells in vivo. Macrophages initiate random cell migration in the 3D environment of a 2–4 h old prepupae[43]. Remarkably, automatic tracking of individual migrating mutant cells revealed a significant increase in cell speed, displacement from origin, and straightness of movement (Fig. 2j, quantification in Fig. 2m, and Supplementary Fig. 2a, b). Control cells migrated with an average track speed mean of 2.3 ± 0.9 μm/min whereas *swip-1* mutant cells migrated considerably faster with an average track speed

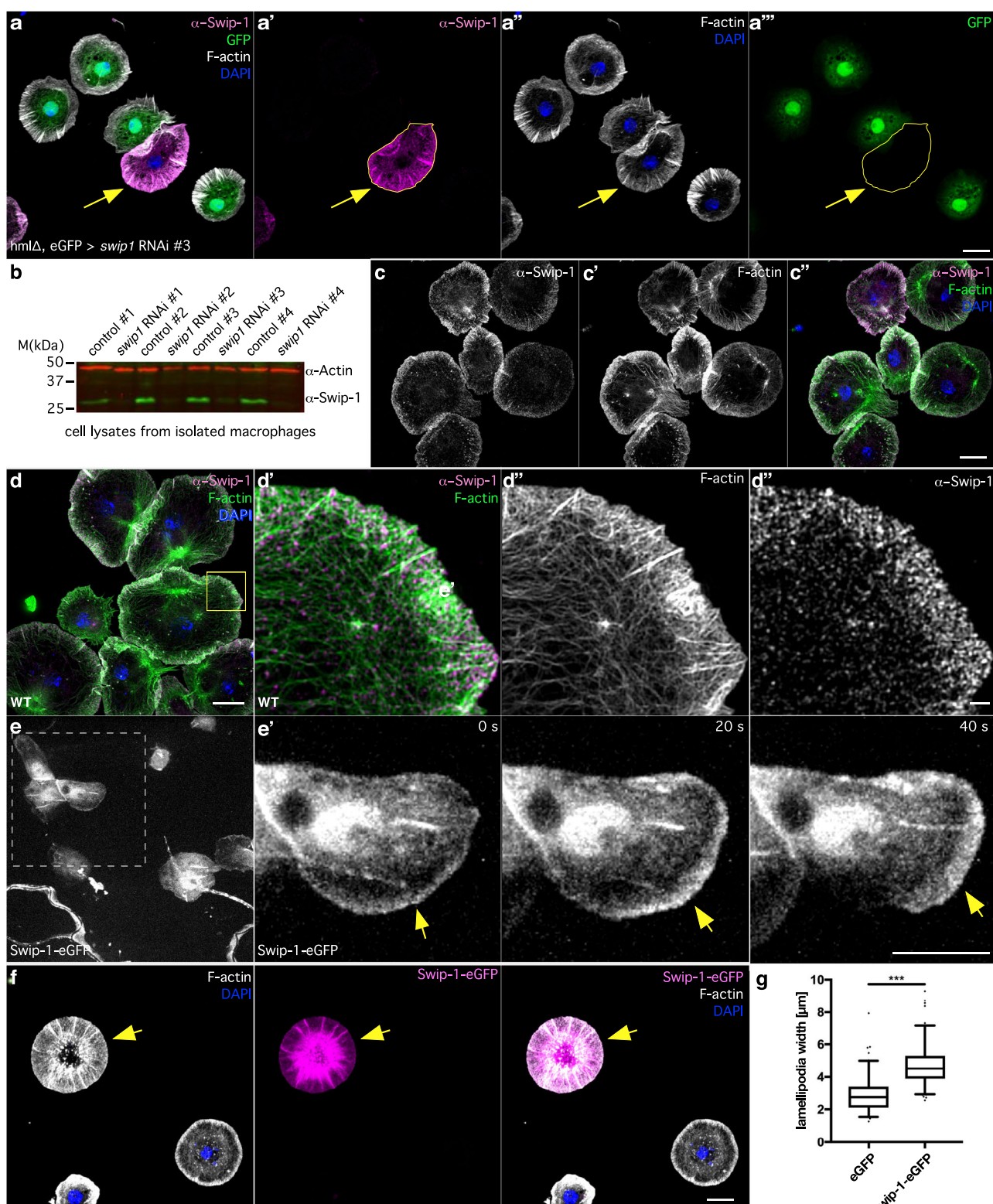

mean of 3.9 ± 0.7 μm/min (Fig. 2m and Supplementary Movie 2). The increased migration speed of mutant cells was rescued by re-expressing Swip-1 (Fig. 2k and quantification in Fig. 2m). By contrast, the overexpression of the same Swip-1 transgene in a wild-type background resulted in subtle but significant reduced migration speed (Fig. 2i and quantification in Fig. 2m). Cells overexpressing Swip-1 formed enlarged but often depolarized lamellipodia protrusions, which is expected to reduce cell

migration speed (Supplementary Movie 2). Thus, these results highlight a conserved role of EFhD2/Swip-1 in lamellipodial protrusions and immune cell migration in vivo.

We next analyzed the functional relevance of calcium binding of Swip-1 in macrophage migration. Based on sequence alignment, *Drosophila* Swip-1 has two predicted EF-hand motifs, EF-loop 1, 82aa <u>DTARDGFLDLQE</u> 93aa (corresponding to human EFhD2 105-116aa), and EF-loop 2, 118aa <u>DEDNDGKISFRE</u> 129aa

**Fig. 1 Swip-1 localizes to protruding lamellipodia. a** Maximum intensity projection of confocal images of the actin cytoskeleton in prepupal macrophages expressing *swip-1* dsRNA marked by eGFP co-expression. Scale bars represent 10 µm. Cells were co-stained with anti-Swip-1 antibody (magenta), DAPI (blue), and phalloidin (white). The arrow marks an eGFP-negative wild-type cell with high Swip-1 expression. **b** Western blot analysis of lysates of macrophages cells expressing different *swip-1* dsRNA transgenes as indicated under the control of the *hemolectin*-Gal4 driver. The expression of dsRNA transgenes without the Gal4 driver serves as a control. Knock-down efficiency was validated with an anti-Swip-1 specific antibody. Actin served as loading control. **c, d** Structured illumination microscopic (SIM) images of wild-type prepupal macrophages co-stained for endogenous Swip-1 (magenta), phalloidin (green) and DAPI (blue). Scale bars 10 µm. **e** Frames of spinning disk microscopic video of a migrating macrophage expressing an Swip-1-eGFP fusion. The arrow marks the localization of Swip-1 in the protruding lamellipodium. Images were taken every 20 s for 30 min (see also Supplementary Movie 1). Scale bar 10 µm. **f** Maximum intensity projection of the confocal image of S2R+ cells transfected with Swip-1-eGFP fusion (magenta), stained with DAPI (blue) and phalloidin (white). Scale bar 10 µm. **g** Lamellipodia width of S2R+ cells either transfected with cytoplasmic eGFP or Swip-1-eGFP fusion was measured at five regions using Image J and averaged. Boxes indicate 50% (25–75%) and whiskers (5–95%) of all measurements, with black lines depicting the medians. n = eGFP WT: 20, Swip-1-eGFP: 21 cells each from one representative transfection. Two-sided Mann–Whitney test was used, P value: <0.001 (***). All images shown are representative of at least three independent experiments.

(corresponding to human EFhD2 141-152aa). Mutations in the first highly conserved aspartate residues of both EF-loops (D82A/D118A; Supplementary Fig. S1c, marked by red boxes) have been shown to abolish calcium-binding capacity[29,44]. Interestingly, re-expression of such mutant Swip-1-D82A/D118A transgene failed to rescue migration defects of *swip-1*-deficient macrophages (Fig. 2I). Remarkably, Swip-1-D82A/D118A mutant protein was no longer enriched in actively protruding lamellipodia, but rather re-localized to the cell cortex suggesting that calcium or full activity might be critical for lamellipodial localization of Swip-1 (Fig. 2n and Supplementary Movie 3). Moreover, mutant macrophages re-expressing Swip-1-D82A/D118A often attached together and formed clusters (Fig. 2n). Cell cluster formation was also prominent in isolated mutant macrophages re-expressing Swip-1-D82A/D118A plated on surfaces coated with the ECM protein vitronectin and correlated with an increased ß-integrin accumulation at cell-cell contacts (Supplementary Fig. 2d). Compared to wild type, mutant cells showed an enhanced spreading and exhibited increased ß-integrin-marked focal adhesion sites along the leading edge (Supplementary Fig. 2e, f) suggesting a possible conserved role of Swip-1 in regulating integrin adhesion dynamics as recently reported in human breast cancer cell line[32].

**Calcium promotes transient F-actin cross-links by EFhD2/Swip-1.** We reasoned that impaired lamellipodia formation in *swip-1* mutant cells could reflect a direct effect of Swip-1 on the structure of the actin cytoskeleton network. Published work suggested that human EFhD2 induces actin bundling in the presence of calcium. This notwithstanding, in this study very high and thus rather non-physiological concentrations of recombinant human EFhD2 protein (up to 15 µM) have been used in in vitro co-sedimentation assays[30]. Moreover, previous binding assays were performed with GST-tagged fusion proteins that noticeably increased F-actin bundling activity of EFhD2[30]. Thus, we first determined the cellular concentration of endogenous Swip-1 in *Drosophila* cells by titrating defined amounts of recombinant protein with total cell lysates from *Drosophila* S2 and S2R+cells in immunoblots. From densitometric intensities, we calculated a cytoplasmic concentration of Swip-1 of about 0.3 µM. We next tested the actin-binding activity of purified recombinant Swip-1 protein in vitro. To exclude artificial oligomerization by the GST moiety, we removed the tag by proteolytic cleavage followed by a final polishing step of the Swip-1 protein. Then, we used high-speed sedimentation to determine the affinity and stoichiometry of Swip-1 to actin (Fig. 3a–d). In the absence of calcium, Swip-1 bound to F-actin with an estimated $K_D$ of $4.3 \pm 1.3$ µM with a ~1:1 (Swip-1: actin) stoichiometry at saturation (Fig. 3a, b). Remarkably, in the presence of calcium, the binding affinity of Swip-1 to F-actin increased approximately five-fold ($K_D$ of

$0.9 \pm 0.1$ µM; Fig. 3c, d). Interestingly, however, the binding stoichiometry changed to ~0.5:1 (Swip-1: actin) in the presence of calcium supporting the notion that Swip-1 undergoes conformational changes upon $Ca^{2+}$ binding[45]. Furthermore, we found in low-speed pelleting assays that in the absence of calcium Swip-1 significantly increased the amount of bundled/cross-linked actin filaments compared to low-speed pelleting assays in the presence of 1 mM $Ca^{2+}$ (Fig. 3e, f and quantification in Fig. 3g).

To explore the effect of Swip-1 on actin filament architecture in more detail, we performed total internal reflection fluorescence (TIRF) microscopy that allows direct visualization of the assembly of single actin filaments and higher-order structures in real time[46]. We first compared actin assembly in the presence of either *Drosophila* Swip-1, human EFhD2, human α-actinin4, or human fascin in the absence of calcium to assess the general behavior of the proteins in vitro (Supplementary Movie 4). In contrast to human proteins, which induced the formation of prominent bundles at 0.5 µM concentrations, we could not observe any formation of actin bundles in the presence of 1.0 µM Swip-1 (Supplementary Movie 5). Rather, under all tested conditions ranging from 0.1–1.0 µM Swip-1 induced exclusively the formation of prominent actin cross-links in the absence of calcium (Fig. 3h and Supplementary Movie 5). Interestingly, we found that human EFhD2 protein exhibited a dual activity in vitro by preferentially cross-linking filaments into loose networks at lower concentration (100 nM) or laterally into loose bundles of filaments at higher concentration (1 µM). Notably, the bundling activity of human EFhD2 was calcium-insensitive but was dominated at higher protein concentrations (Supplementary Movie 6). In stark contrast, the actin filament cross-linking activity of *Drosophila* Swip-1 (Supplementary Movie 7) and human EFhD2 (Supplementary Movie 8) is regulated by calcium. Quantitative analysis of cross-link events revealed that in the absence of calcium even at 0.1 µM, Swip-1 induced stable, perpendicular cross-links between elongating actin filaments, creating a stable and highly cross-linked network over time (Fig. 3h and Supplementary Movie 5). However, cross-links between actin filaments were rare or nearly absent in the presence of calcium at the low Swip-1 concentration of 0.1 µM. At 1.0 µM Swip-1, the cross-links were more frequent, but still highly dynamic and short-lived as assessed by quantification of attachment versus detachment events of cross-linked filaments from time-lapse TIRF movies (Fig. 3h, i and Supplementary Movie 7). Similarly, we observed that human EFhD2/Swip-1 frequently induced stable cross-links in the absence of calcium, while it only bundled actin filaments in the presence of calcium (Supplementary Movie 8).

To assess the functional relevance of calcium binding of Swip-1 on actin filament architecture in vitro, we purified a recombinant

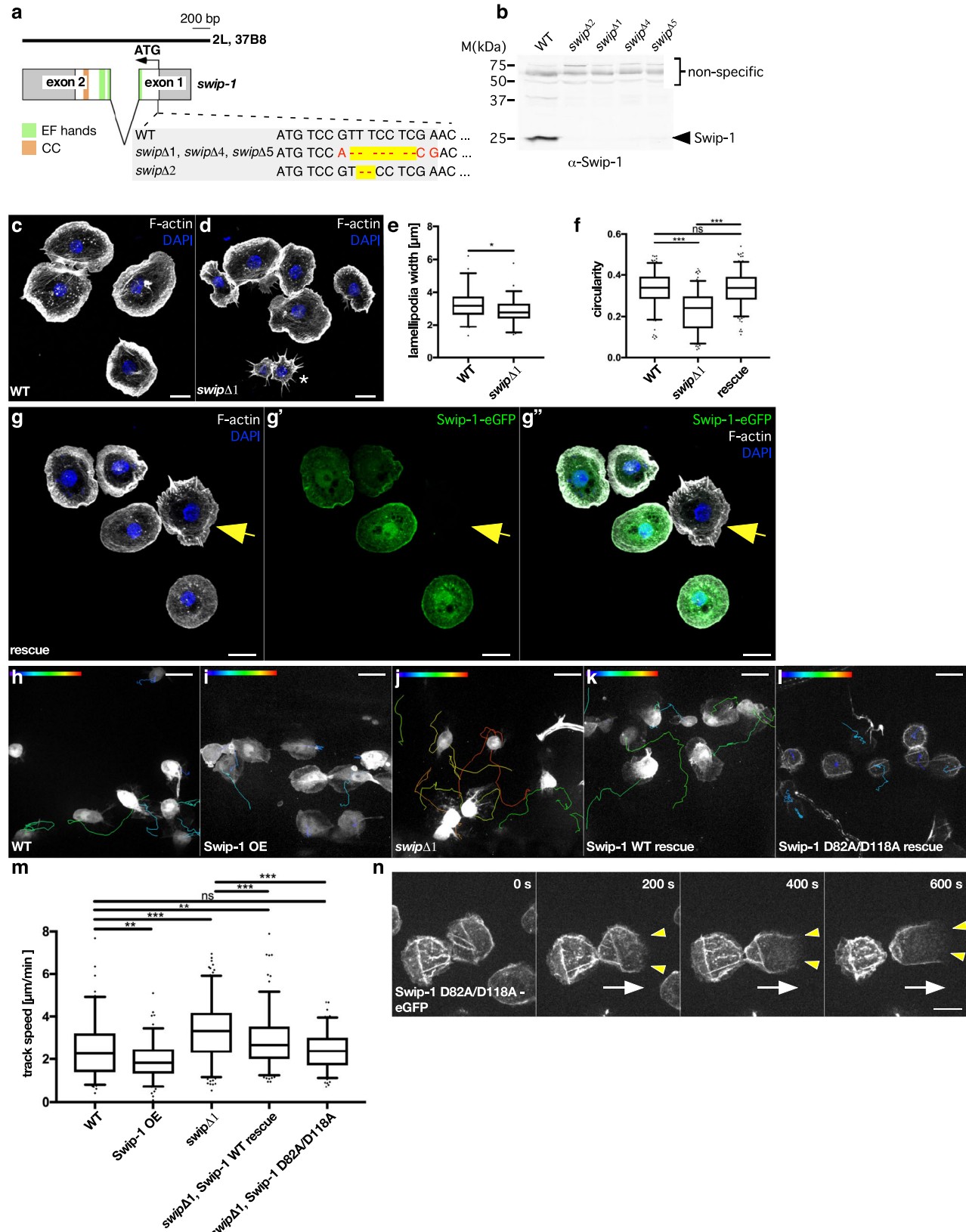

calcium-binding deficient Swip-1 mutant protein D82A/D118A in which the highly conserved first coordinating aspartic acid of both EF-loops was replaced with an alanine. Subsequent TIRF analysis revealed that this mutant protein still induced stable, perpendicular cross-links, albeit in a calcium-independent manner (Supplementary Fig. 3 and Supplementary Movie 9).

Unexpectedly, however, this mutant Swip-1 protein additionally acquired prominent actin-bundling activity (Supplementary Movie 9), suggesting a conformational change in the protein, caused by the exchange of the two negatively charged aspartic acid residues with alanine. Notably, conformational changes in the EF-hand triggered by calcium binding have been previously

**Fig. 2 Loss of Swip-1 impairs lamellipodia formation and cell migration of macrophages. a** Schematic overview of the *swip-1* gene locus. Exons encoding parts of distinct domains are indicated, EF domains in green and coiled-coil region in orange. The target sequence for CRISPR/Cas9 gene modification and generated *swip-1* deletions are depicted. **b** Loss of *swip-1* mutants were validated by Western blot analysis. Lysates from ten fly heads of wild-type and different mutant flies were analyzed. Non-specific bands (upper bands) serve as a loading control. All obtained mutants were validated once, hereinafter used swipΔ1 mutant flies were checked regularly in Western blot analysis of head lysates. **c**, **d** Confocal images of **c** wild-type and **d** *swip-1* mutant macrophages were co-stained with phalloidin (white) and DAPI (blue). Scale bars 10 μm. **e** Quantification of lamellipodia width at the leading edge, *n* = WT: 64, *swip-1* mutant: 61 cells of two independent experiments (*P* value: 0.012). **f** Circularity was measured using Image J shape descriptors, *n* = WT: 180, mutant: 192 (*P* value: >0.001), rescue: 209 cells (*P* value: 0.892). **g** Rescue of *swip-1* mutant macrophages re-expressing a Swip-1-eGFP fusion were co-stained with phalloidin (white) and DAPI (blue). The arrow marks mutant cells lacking Swip-1-eGFP that still show an irregular cell shape. **h**, **i** Spinning disk microscopy live imaging of randomly migrating macrophages in prepupae with indicated genotypes were tracked by using Imaris 9.3. Track speed mean is color-coded (0.06 to 6 μm/min) (**h**) wild type and (**i**) overexpression of a Swip-1-eGFP (OE) in a wild-type background, **j** *swip-1* mutant, (**k**) rescue with a Swip-1-eGFP, and **l** rescue with a Swip-1-D82A/D118A-eGFP, respectively. **m** Quantification of track speed mean, WT: *n* = 114 tracks, OE: *n* = 133 tracks (*P* value: 0.006), *swip-1* mutant: *n* = 206 tracks (*P* value: <0.001), rescue WT: *n* = 166 tracks (*P* values: 0.002 to WT and <0.001 to mutant), rescue SwipD82A/D118A: *n* = 136 tracks (*P* values: 0.594 to WT and <0.001 to mutant). **e**, **f**, **m** Boxes indicate 50% (25–75%) and whiskers (5–95%) of all measurements, with black lines depicting the medians. For statistical analysis, two values indicated by connecting black lines were compared with two-sided Mann–Whitney test, *P* value: 0.12 (ns), 0.033 (*), 0.002 (**), <0.001 (***). **n** Frames of spinning disk microscopy video of migrating *swip-1* mutant macrophages re-expressing Swip-1-D82A/D118A-eGFP. Images were taken every 20 s for 30 min. A white arrow indicates the direction of movement; yellow arrowheads mark the lamellipodium. Swip-1-D82A/D118A-eGFP localizes to the cell cortex but not to the protruding lamellipodium (see also Supplementary Video M3). Scale bar 10 μm. All images shown are representative of at least three independent experiments unless otherwise specified.

also shown to modulate F-actin binding in both human α-actinin and EFhD2[45,47]. Taken together, our data uncover a calcium-regulated actin cross-linking activity of EFhD2/Swip-1, conserved between humans and flies, which might be relevant in diverse cellular processes depending on calcium-induced rapid reorganization of dynamic actin networks.

**Swip-1 localizes to lamellipodial protrusions in a calcium-dependent manner during epithelial wound closure.** To better understand the physiological relevance of Swip-1 function within a calcium-dependent in vivo tissue context, we established a single-cell wounding model system using larval epidermal cells (LECs) in the early pupal stages of *Drosophila* development. LECs are up to 70 μm in diameter cells and form a polarized epithelial monolayer. During mid-metamorphosis LECs undergo apoptosis and are replaced by epidermal histoblast cells forming the new adult epidermal sheet[48].

Spinning disk, live-cell imaging microscopy of the dorsal side of the abdomen of 18 h APF old pupae ubiquitously expressing a LifeAct-eGFP transgene revealed a tight epidermal sheet consisting of large polygonal epithelial cells (Fig. 4a, b). Epithelial cells showed an overall stationary cell behavior with only very small membrane protrusions formed along their cell–cell contacts (Fig. 4b; unwounded). This was dramatically changed upon laser-induced single-cell wounding. In the first minutes, F-actin assembled into broad lamellipodial protrusions within cells at the wound edge (front row; Fig. 4c and Supplementary Movie 10). Time-lapse movies further demonstrated that lamellipodial protrusions reached a maximum size between 5–10 min after wounding. Thereafter, lamellipodia formation decreased and coexisted with an increasing number of contractile actin bundles, which were formed at the leading edge of the wound, contracted laterally to pull cells forward, and increasingly contributed to wound closure (Fig. 4c and Supplementary Movie 10).

Imaging of a Swip-1-eGFP tagged transgene immediately after laser cell ablation revealed prominent recruitment of Swip-1 from the cell periphery into newly formed lamellipodial protrusions oriented towards the wound area (Fig. 4d and Supplementary Movie 11). Remarkably, we observed that after some time, even epithelial cells several rows back from the wound site exhibited a delayed response by extending lamellipodia marked by EFhD2/Swip-1-eGFP along intact cell–cell contacts (marked by white arrows in Fig. 4d and Supplementary Movie 11). This could also

be observed in epithelial wounds using the Lifeact-eGFP transgene (Supplementary Movie 10).

This suggests that the initial wound signal is transmitted to undamaged cells, possibly by a rapid influx of calcium into the cells as previously observed in embryonic and pupal wound models[49,50]. To test this hypothesis, we took advantage of an mRuby-based RCaMP1, a genetically encoded calcium reporter[51]. Expression of RCaMP1 under the control of *da*-GAL4 did not reveal significant changes in calcium levels before wounding (Fig. 4e and Supplementary Movie 12). However, immediately after wounding, a dramatic increase of intracellular calcium was observed as a bright fluorescence signal, first observed in front-row cells, but then rapidly spreading to more distal cell rows within 20 s (Fig. 4e and Supplementary Movie 12). After initial calcium wave propagation, the levels of intracellular calcium decreased from the periphery to the margin of the wound within a minute (Supplementary Movie 12). Combined, these data strongly suggest that wound-induced cytosolic calcium increase might promote transient remodeling of the cortical actin cytoskeleton to allow subsequent lamellipodial protrusion and cell constriction driving wound closure (Fig. 4f).

**Swip-1 function is required for epithelial wound closure in vivo.** We then explored the physiological function of Swip-1 in epithelial wound closure in more detail. Epithelial cells deficient for Swip-1 showed prominent wound closure defects (Fig. 5b, c). Mutant cells at the wound edge still formed cell protrusions (Fig. 5b, yellow arrowheads). However, the size and the sheet-like shape of these protrusions were reduced with more prominent irregular, highly unstable spike-like filopodia unlike wild-type cells (Supplementary Movie 13). To better quantify wound closure defects, we measured the wound area over 60 min, normalized to the initial wound size at time zero (Fig. 5c). In the wild type, in the first 90 s after ablation, the wound area first expanded as a possible consequence of tissue tension release as previously reported in embryonic wound closure[49,50]. Thereafter, wild-type cells immediately initiated the formation of lamellipodial protrusions (see Figs. 4c and 5c). By contrast, in *swip-1* deficient cells the initial expansion phase was prolonged and the formation of protrusions was markedly delayed (Fig. 5b, c). Even greater differences were observed in the subsequent contraction of cells at the wound margin, which already started after five minutes after ablation in the wild type. In contrast, in *swip-1* mutants wounds either completely failed to constrict or showed a dramatic delay

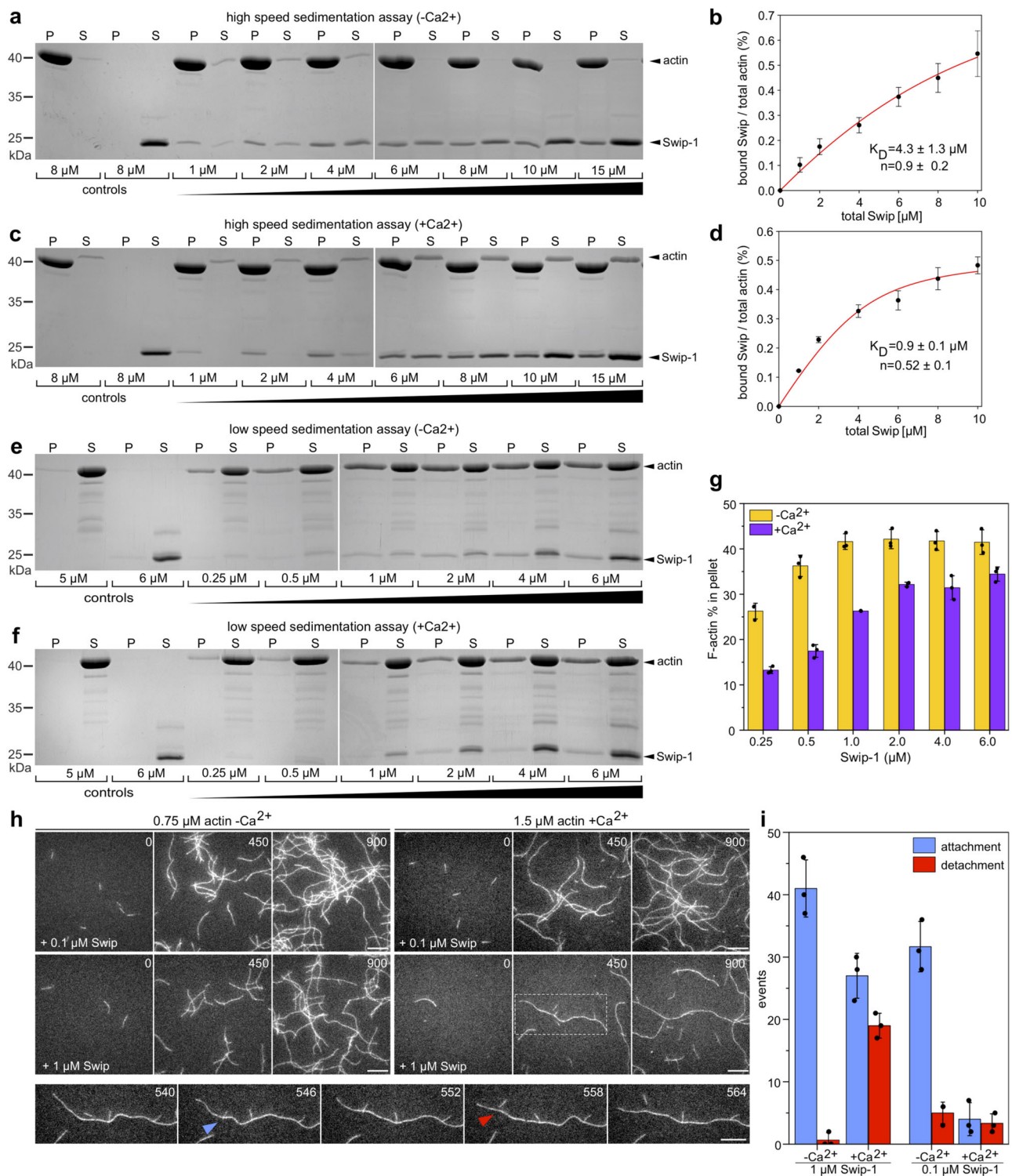

up to 25 min. Accordingly, mutant wounds remained nearly open even at 60 min after ablation, whereas wild-type wounds were generally already closed up to 70% of the initial wound size (Fig. 5d). Re-expression of a full-length Swip-1 protein, but not deletion constructs lacking either the EF-hands (ΔEF), deficient for calcium binding (EFhD2/Swip-1-D82A/D118A) or the coiled-coil domain (ΔCC) could substantially rescue wound closure defects (Fig. 5c, d). Taken together, these data indicate that appropriate wound closure requires both functional calcium-binding and dimerization of Swip-1. Mechanistically, these data provide a mechanism for how elevated calcium levels result in

rapid reorganization of actin networks driving epithelial wound closure. Given the cryptic lamellipodia formed at the mutant epithelial wound edge, Swip-1 seems to be not only essential for the initial calcium-dependent remodeling of actin filaments, but also later for stabilizing lamellipodia protrusions when calcium has been decreased to basal levels.

**EFhD2/Swip-1 stabilizes lamellipodial protrusions for efficient force transmission in adhesive 2D cell migration.** To experimentally test the relevance of EFhD2/Swip-1 in lamellipodium

**Fig. 3 EFhD2/Swip-1 is a calcium (Ca$^{2+}$)-regulated cross-linking protein. a, c** Co-sedimentation of *Drosophila* Swip-1 with F-actin in high-speed pelleting assays in the absence or presence of 1 mM Ca$^{2+}$. Increasing concentrations of *Drosophila* Swip-1 as indicated were incubated with 8 μM G-actin in polymerization buffer, and the proteins recovered in the pellet (P) and the supernatant (S) fractions after the centrifugation at 200,000 × *g* were stained with Coomassie blue. **b, d** Quantification of equilibrium constants and binding sites (*n*) on F-actin in the absence or presence of Ca$^{2+}$ from experiments as shown in **a, c**. Points represent mean ± SD of increasing concentrations from three independent co-sedimentation experiments (*n* = 3). Solid red lines represent calculated binding isotherms. **e, f** Co-sedimentation of *Drosophila* Swip-1 with F-actin in low-speed pelleting assays at 20,000 × *g* in the absence or presence of 1 mM Ca$^{2+}$. Increasing concentrations of Swip-1 as indicated were incubated with 5 μM G-actin in polymerization buffer, and the proteins in the pellet (P) and the supernatant (S) fractions were stained with Coomassie blue. **g** Quantification of F-actin in pellet fractions from experiments as shown in **e, f**. Bars represent mean ± SD of three independent co-sedimentation experiments (*n* = 3). Data points show the measured values of the experiments. **h** Ca$^{2+}$ prevents stable cross-linking of actin filaments caused by *Drosophila* Swip-1. Time-lapse micrographs of TIRFM assays, used for the analysis of cross-linking behavior of Swip-1 in the absence or presence of 1 mM Ca$^{2+}$. Polymerization of 0.75 or 1.5 μM G-actin (10% ATTO488-labeled) with 0.1 μM (top) and 1.0 μM *Drosophila* Swip-1 (middle) in the absence and presence of Ca$^{2+}$. Note significantly cross-linked networks in the absence of Ca$^{2+}$ as compared to assay conditions in the presence of Ca$^{2+}$. The enlarged gallery of inset (dashed box) at the bottom displays dynamic cross-linking behavior of 1.0 μM *Drosophila* Swip-1 in the presence of Ca$^{2+}$ at higher temporal resolution. The blue arrowhead marks an attachment event of a short filament with the longer filament and the red arrowhead depicts the detachment of the short filament after 12 s. Time is given in seconds in the upper right corner of each frame. Scale bars 10 μm. **i** Quantification of attachment and detachment events in the absence or presence of Ca$^{2+}$. Bars represent mean ± SD from three movies each (*n* = 3). Data points show the measured values of the experiments.

formation and protrusion in the absence of calcium gradients, we analyzed the function of EFhD2 in the highly polarized B16-F1 mouse melanoma cells forming prominent lamellipodia when migrating on laminin on a 2D surface. Subcellular localization of human EFhD2 in transiently transfected B16-F1 mouse melanoma cells confirmed its conserved localization in lamellipodial protrusions (ref. [30] and Fig. 6a). Human EFhD2 fused to EGFP could readily be observed in F-actin enriched lamellipodial protrusions and on intracellular vesicles (Fig. 6a). Co-immunostaining with antibodies directed against the WRC subunit WAVE2, the actin polymerase VASP, and the F-actin binding protein cortactin, which is highly reminiscent of Arp2/3 localization, revealed that EFhD2 localizes broadly within lamellipodia but not to the tip of the leading edge of B16-F1 cells (Fig. 6a). The siRNA or shRNA-mediated knockdowns of EFhD2 in HeLa, NIH 3T3, and B16-F10 cells have been previously reported to diminish cell spreading and wound healing in the scratch assay[30,41]. The latter study additionally observed diminished cell speed of EFhD2-depleted B16-F10 cells stably transduced with shRNA EFhD2 and shRNA-control in 2D assays. To thoroughly examine the function of EFhD2 in lamellipodium formation, protrusion, and 2D-cell migration on the single-cell level in a genetic knockout, we inactivated EFhD2 by CRISPR/Cas9 in B16-F1 cells. Disruption of the *EFhD2* gene in two independent clonal cell lines (KO #10 and #33) was verified by sequencing of genomic target sites and loss of protein was additionally confirmed by immunoblotting (Fig. 6b). Migration rates of B16-F1 wild-type cells and both mutants were then analyzed on laminin by phase-contrast time-lapse microscopy (Supplementary Movie 14). In line with the previous work[41], loss of EFhD2 in B16-F1 cells caused a significant reduction in cell speed (Fig. 6c). Moreover, effective migration as assessed by quantification of mean square displacement (MSD) was substantially diminished (Fig. 6d). Since 2D migration on flat surfaces is largely driven by actin assembly in the lamellipodium, we then analyzed actin filament (F-actin) content in B16-F1 and EFhD2-KO cells after phalloidin staining. Notably, quantitative analysis of lamellipodia revealed no changes in F-actin intensity (Fig. 6f). Interestingly, however, lamellipodia widths in EFhD2 knockout cells on 2D substrates were reduced by almost 40% as compared to control (Fig. 6g). The number of microspikes in mutant cells though, remained unchanged as compared to control (Fig. 6h). Finally, we asked whether or to which extent lamellipodia protrusion was affected. To this end, we recorded the wild type and EFhD2-deficient cells randomly migrating on laminin by time-lapse, phase-contrast microscopy and determined the respective protrusion rates by kymograph analyses (Fig. 6i, j). Quantification

revealed lamellipodia protrusion to be reduced by about 30% in both mutants compared to the B16-F1 control (Fig. 6k). Combined these data strongly suggest that cross-linking of adjacent actin filaments within dendritic arrays by EFhD2 significantly contributes to the mechanical stability of lamellipodia to drive efficient protrusion.

## Discussion

**EFHD2/Swip-1 has a conserved function in lamellipodia-based protrusion and cell migration**. Previous studies have implicated EFhD2/Swip-1 in a number of cellular processes ranging from adhesion turnover, cell spreading and migration, B cell receptor signaling, and cancer invasion[52]. In this study, we identified an important role of EFHD2/Swip-1 in controlling lamellipodial protrusions and cell migration of *Drosophila* immune cells and B16-F1 mouse melanoma cells. How does EFhD2/Swip-1 act on lamellipodia-based protrusion? In variance with previous work, we found that human EFhD2 not only bundles actin filaments but also efficiently cross-link actin networks in vitro comparable to *Drosophila* Swip-1. The actin cross-linking activity of EFhD2 synergizes with WRC-Arp2/3-branched actin nucleation promoting the generation of a stable and densely branched actin filament network (Supplementary Movie 15). Supporting this notion, we found that loss of EFhD2/Swip-1 in both flies and B16-F1 cells decreases lamellipodia widths associated with markedly diminished protrusion rates, whereas forced over-expression of EFhD2/Swip-1 increases lamellipodia widths when plated on 2D surfaces. Given that cross-linking proteins largely determine the structure and viscoelastic properties of F-actin networks[53], our findings strongly suggest that EFhD2/Swip-1 tunes mechanical force generation in protruding lamellipodia.

However, different from migrating B16-F1 cells in vitro, *Drosophila* macrophages deficient for Swip-1 show a significantly increased migratory speed in vivo. Notably, a similar observation was recently made in EFhD2-depleted mice in which B-cells were also shown to migrate faster in vivo[35]. This could be due to different protein requirements as well as different physical requirements for the distinct migration modes. The architecture of the actin cytoskeleton also depends on substrate rigidity and cells adapt to different extracellular resistance by reorganizing their actin network[54,55]. Thus, without cross-linking activity upon loss of EFhD2, the Arp2/3 complex might be insufficient for maintaining the mechanical stability of the branched actin network at the leading edge of B16-F1 cells. As a consequence, loss of EFhD2 in B16-F1 cells caused a significant reduction in

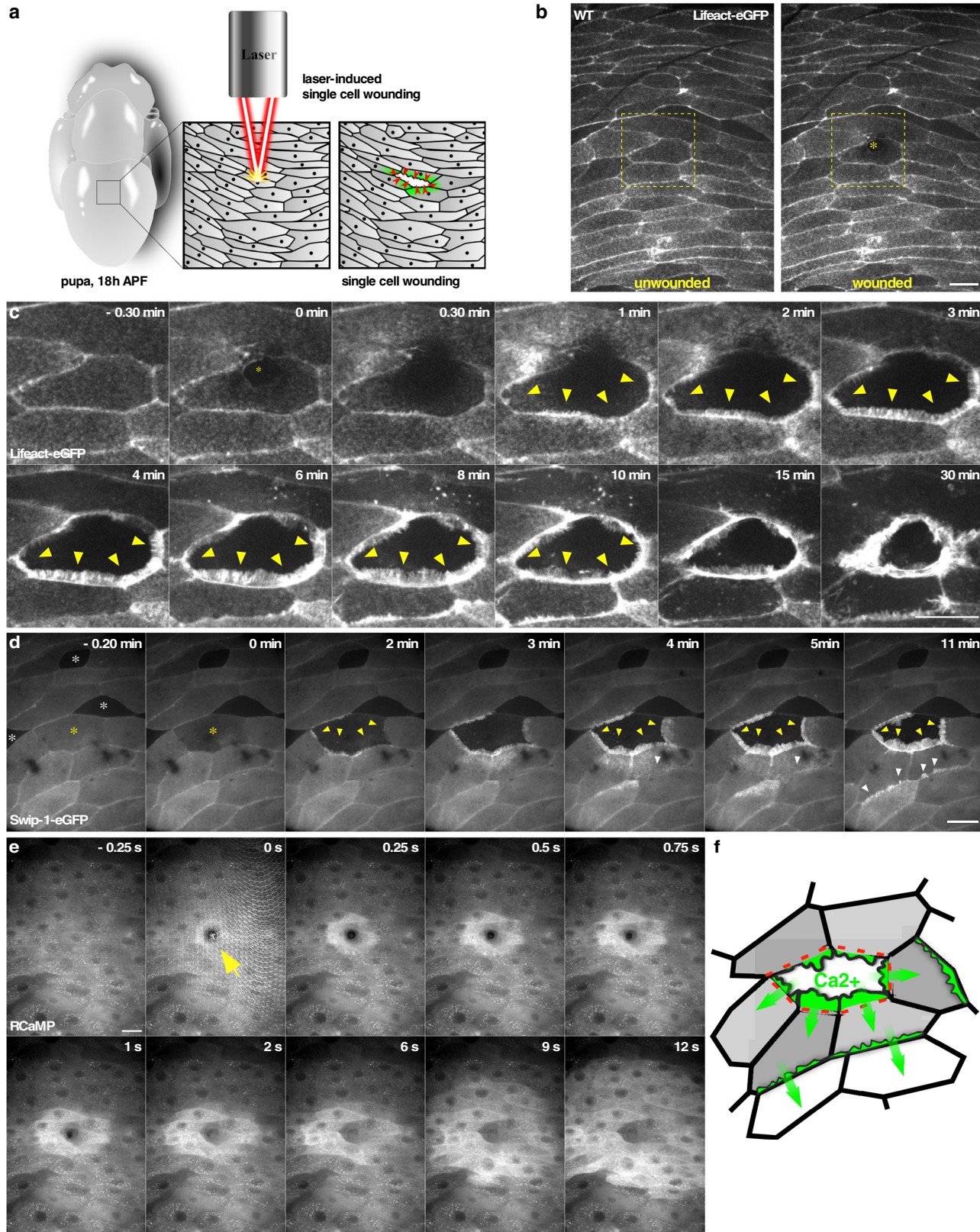

lamellipodial protrusion, cell speed, and effective migration. By contrast, immune cells migrate across a complex 3D environment with highly varying extracellular resistance that requires more dynamic protrusions. Thus, the loss of Swip-1 in *Drosophila* macrophages could result in a reduced actin network density that could lead to faster but less persistent migration in vivo.

Supporting this notion, we found that forced overexpression of Swip-1 stabilizes protrusions at the expense of migration speed in vivo. Thus, the assembly of cross-linking proteins such as EFhD2/Swip-1 into the branched actin network might contribute to a resistance-adaptive behavior of migrating cells as recently proposed[56].

**Fig. 4 Swip-1 is recruited to lamellipodial protrusions during epithelial wound closure in vivo. a** Schematic of the in vivo model to study calcium-dependent wound healing in early *Drosophila* pupal stages. **b, c** Single-cell ablation in the abdominal epidermis of a wild-type 18 h APF old pupa ubiquitously expressing a Lifeact-eGFP transgene under the control of the *da*-Gal4 driver. Images were taken every 30 s for 30 min, ablation starts at $t = 0$ min. **b** Overview of the imaged area of the monolayered epithelium, an asterisk indicates ablated cell. Scale bar 25 μm. **c** Magnification of the ablated cell of **b** at the indicated times. Arrows show forming lamellipodial protrusions at the wound margin. Scale bar 25 μm. **d** Single-cell ablation in the abdominal epithelium of a 18 h APF old pupa ubiquitously expressing Swip-1-eGFP transgene; yellow asterisk indicates the position of the ablated cell. White asterisks mark specialized epidermal cells, so-called tendon cells providing attachment sides to the muscles. These tendon cells lack Swip-1-eGFP expression driven by the *da*-Gal4 driver. Yellow arrows mark lamellipodial protrusions at the wound margin, whereas white arrows mark induced lamellipodia of cells several rows back from the wound site exhibited a delayed response. Scale bar 25 μm. **e** Single-cell ablation in the abdominal epidermis of a 18 h APF old pupa ubiquitously expressing the calcium indicator RCaMP. Changes of intracellular calcium are visible as increasing fluorescence intensity. The calcium wave is propagated in a circular fashion with a velocity of 2.2 ± 0.7 μm/s. Scale bar 25 μm. **c–e** Images shown are representative of **a** nine and **b, c** at least three independent experiments. **f** Schematic of the calcium wave-inducing membrane protrusions after wounding.

In addition, EFhD2/Swip-1 has been recently identified as a cargo-specific adapter for CG-endocytosis controlling integrin adhesion turnover[32]. This function might be also conserved and could further contribute to the changed migratory behavior of macrophages in vivo. Interestingly, we found that *swip-1*-deficient macrophages and those re-expressing the calcium-binding mutant Swip-1-D82A/D118A variant showed an increase in integrin-marked adhesions on vitronectin-coated surfaces. Enlarged focal adhesion caused by slower integrin turnover might further contribute to an increase in the migration speed as similarly observed for *zyxin*-deficient macrophages in *Drosophila*[43]. Interestingly, such a prominent inverse relationship between focal adhesion size and cell migration speed has been shown in numerous motile cell types[57]. The coordinated disassembly of integrin-mediated focal adhesions requires $Ca^{2+}$ influx[58,59]; however, whether the function of EFhD2/Swip-1 in integrin traffic depends on its calcium-binding activity has not yet been addressed[32].

**Calcium bursts allow a rapid reorganization of the actin cytoskeleton through Swip-1.** The mechanical properties of the actin cytoskeleton can not only adapt in response to extracellular resistance but also in response to intracellular calcium changes. Closure of wounds entails complex 3D movements of epithelial cells that are orchestrated by calcium signaling[49,60,61]. Laser-induced wounding immediately triggers a rapid calcium flash extending outward from the point of wounding as a wave[49]. Epithelial cells at the wound margin respond by polarizing their actin cytoskeleton to drive both protrusive structures and supracellular contractile actin cables, both contributing to efficient wound closure. During embryonic wound healing, lamellipodial and filopodial protrusions are often induced in later stages promoting the final sealing of the wound[60,62]. In our single-cell wounding model, wound closure starts with the formation of remarkably broad lamellipodial protrusions that depend on Swip-1 function. Swip-1 is immediately recruited to those lamellipodial protrusions not only directly adjacent to the wound but also in cells several rows back from the wound edge. Thus, Swip-1 resides at the heart of the first most prominent morphological changes in wound closure that require dynamic remodeling of the actin filament meshwork. How does Swip-1 contribute to this initial process? Previous in vitro studies reported that calcium promotes the bundling activity of EFhD2[45]. In this study, we could confirm the bundling activity of human EFhD2 protein by TIRF imaging. Contrary to previous assumptions, however, EFhD2-induced actin bundling was found to be calcium-insensitive. More importantly, we identified a conserved cross-linking activity that is regulated by calcium. As increased calcium levels promote the release of existing actin cross-links formed by EFhD2/Swip-1, we suggest a model in which elevated calcium concentrations reduce

EFHD2/Swip-1 cross-linking activity to promote rapid reorganization of existing actin networks, e.g. the cortical actin layer in pupa epithelia, allowing to drive fast and efficient epithelial wound closure by extension of newly formed lamellipodia (Fig. 7). Consistently, re-expression of calcium-binding deficient Swip-1 mutant did not rescue wound closure defects in *swip-1* deficient epidermal cells. As expected, this calcium-binding mutant still induced stable cross-links in a calcium-independent manner. Unexpectedly, however, this mutant protein additionally induced prominent actin bundles, an activity exclusively observed with human EFhD2. Of note, the determined X-ray structure of human EFhD2[45] and the predicted 3D structure of *Drosophila* Swip-1 computed by AlphaFold[63] could be precisely superimposed, thus excluding major conformational differences as the underlying reason explaining differential bundling or cross-linking of actin filament in the absence of calcium. On the other hand, conformational changes in EF-hands triggered by calcium binding, as previously shown for human EFhD2[45], are likely to be responsible for the dissociation of cross-links formed by EFhD2 and Swip-1. We further hypothesize, that the acquired bundling activity of the Swip-1 mutant could also be driven by enhanced electrostatic interactions of the negative actin molecule with Swip-1, in which two negative residues were replaced by alanine (D82A/D118A). High-resolution X-ray structures of wild-type and mutant Swip-1 proteins will be necessary to resolve this important issue in future work.

Defective and delayed constriction of the wound margin in mutant epidermal cells further suggests that Swip-1 function is not only required for the formation and dynamics of lamellipodial protrusions but might also be involved in the subsequent formation of contractile actin cables forming along the wound margin. Supporting this notion, mouse EFhD2 has been recently identified as a binding partner of non-muscle myosin II heavy chain isoforms, including myosin 2a and myosin 2b[64]. Interestingly, non-muscle myosin II is not only a motor protein with contractile functions, but it can also cross-link actin filaments[65,66]. Thus, EFhD2/Swip-1 might act synergistically as a cross-linking protein together with myosin II in actomyosin-based contraction during wound closure, since myosin II has also been found to be immediately recruited to the wound edge after injury[67]. Given the large sizes of post-mitotic LECs, our single-cell wound system allows us to genetically dissect these different phases of wound healing at high spatial and temporal resolution in future work.

## Methods

**Bulk RNA-sequencing and bioinformatic analysis.** Total cellular RNAs from total larval hemocytes and pupal hemocytes, Hml+ larval hemocytes, and Hml+ pupal hemocytes were isolated in triplicates using the TRIzol Reagent (Thermo Fisher) following the manufacturer's protocol. The concentrations of RNA were determined using a NanoDrop ND-1000 (NanoDrop). The quality and integrity of the RNA

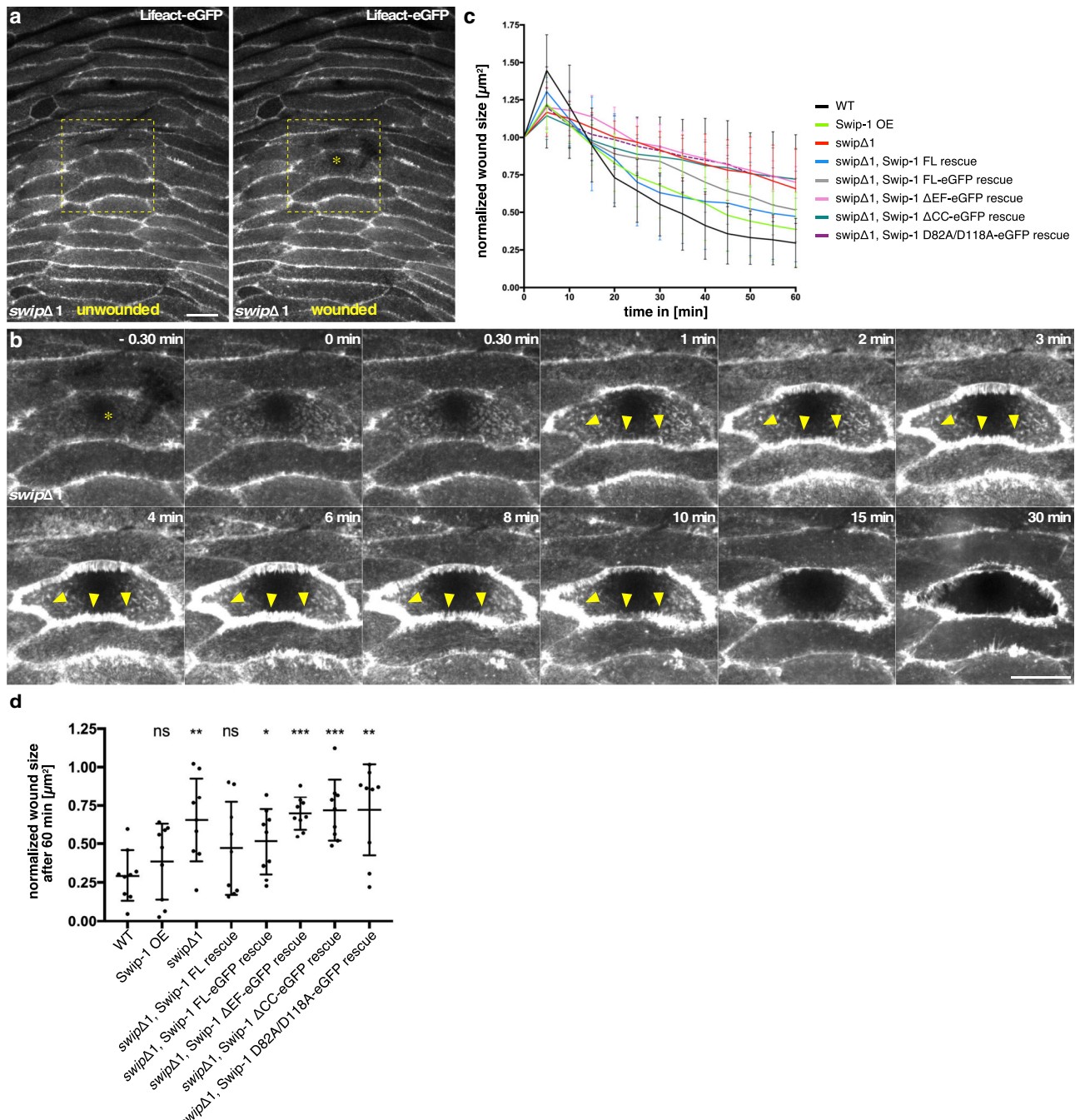

**Fig. 5 Epithelial wound closure requires Swip-1 function. a** Single-cell ablation in the abdominal epidermis of a *swip-1* mutant 18 h APF old pupa ubiquitously expressing a Lifeact-eGFP transgene under the control of the *da*-Gal4 driver. Images were taken every 30 s for 60 min, ablation starts at *t* = 0 min. Overview of the imaged area, an asterisk indicates the position of cell ablation. Scale bar 25 μm. The image is representative of nine independent experiments. **b** Magnification of ablated cell shown in **a** at the indicated times, arrows indicate irregular unstable lamellipodial protrusions at the wound margin. Scale bar 25 μm. **c** Quantification of wound closure in wild type (WT), *swip-1* mutant (swipΔ1), after overexpression (Swip-1 OE) and rescue by re-expressing distinct Swip-1 transgenes, FL: full-length; ΔEF: lacks the EF hands and ΔCC: lacks the coiled-coil region. Wound size was measured every 5 min and normalized to the initial size of the unwounded cell. **d** Excerpt of data from **c**. After 60 min, wound closure was assessed by comparing the remaining wound size normalized to unwounded cell size to wild type. Each point represents one experiment. *P* values: Swip-1 OE: 0.387, swipΔ1: 0.008, swipΔ1, Swip-1 FL rescue: 0.340, swipΔ1, Swip-1 FL-eGFP rescue: 0.040, swipΔ1, Swip-1-ΔEF-eGFP rescue: <0.001, swipΔ1, Swip-1-ΔCC-eGFP rescue: <0.001, swipΔ1, Swip-1-D82A/D118A-eGFP rescue: 0.006. **c**, **d** Bars represent the mean ± SD of nine independent experiments for each genotype. Two-sided Mann–Whitney test was used to compare each genotype individually to WT, *P* values: 0.12 (ns), 0.033 (*), 0.002 (**), <0.001 (***).

were assessed with the Fragment Analyzer (Advanced Analytical) by using the standard sensitivity RNA Analysis Kit (DNF-471). All samples selected for sequencing exhibited an RNA integrity number over 8. RNA-seq libraries were prepared from 100 ng total RNA using the TruSeq RNA Library Prep Kit (Illumina). For accurate quantitation of libraries, the QuantiFluor® dsDNA System (Promega) was used. The fragment size of each library was analyzed using a Fragment Analyzer (Advanced Bioanalytical) by using the high sensitivity RNA Kit (Agilent). Libraries were pooled and sequenced on the Illumina HiSeq 4000 (SE; 1 × 50 bp; 20–30 Mio reads/sample). Demultiplexing was done by using bcl2fastq v2.17.1.14. Adapter sequences were removed from demultiplexed fastq files by trimmomatic and

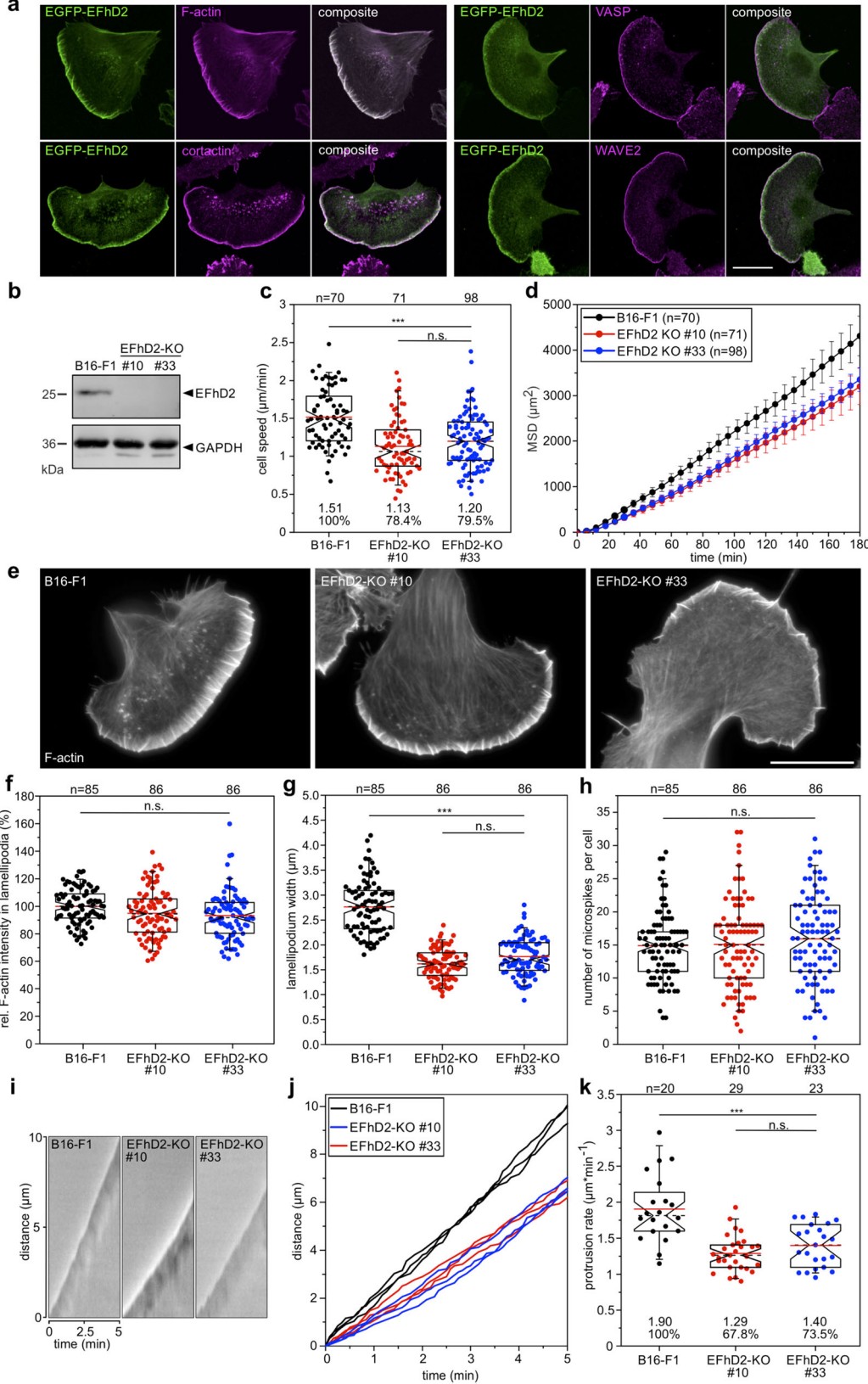

sequence quality was assured via FastQC. The reads were aligned to a reference Drosophila genome (dm6-v101) (Drosophila_melanogaster.BDGP6.28.101.gtf) using STAR aligner 2.5[68]. Differential gene expression analysis was performed with DESeq2 v1.26.0[69]. The difference in expression patterns among samples was analyzed by principal component analysis (PCA). Visualization of differentially

expressed genes was performed with the heatmap package in Rstudio Version 1.4.1103.

**Drosophila genetics**. Fly husbandry and crossing were carried out according to the standard methods. All crosses were performed at 29 °C. The following fly stocks

**Fig. 6 Loss of EFhD2 impairs lamellipodia formation and adhesive 2D cell migration of mouse B16-F1 cells. a** EGFP-tagged EFhD2 expressed in B16-F1 cells co-localizes prominently with F-actin in the entire lamellipodium including microspikes. The other images display EGFP-EFhD2 expressing cells additionally stained for the actin polymerase VASP, cortactin, and the Arp2/3 complex activator WAVE2. Scale bar 20 µm. **b** Elimination of EFhD2 by CRISPR/Cas9 in two independent B16-F1 mutants (#10 and #33) was confirmed by immunoblotting using specific antibodies. GAPDH was used as a loading control. **c** Elimination of EFhD2 diminishes cell migration on laminin. Three time-lapse movies from three independent experiments were analyzed for each cell line, B16-F1: $n = 70$ cells tracked, mutant#10: $n = 71$ cells tracked ($P$ value: <0.001 to B16-F1), mutant#33: $n = 98$ cells tracked ($P$ value: <0.001 to B16-F1 and 0.671 to mutant#10). **d** Analyses of mean square displacement of wild-type versus mutant cells. Respective symbols and error bars represent the means ± SEM of three time-lapse movies from three independent experiments ($n = 3$). **e** Loss of EFhD2 perturbs lamellipodia formation in B16-F1 cells. Representative examples of lamellipodia from wild-type B16-F1 and EFhD2-KO mutant cells. Cells migrating on laminin were stained for the actin cytoskeleton with phalloidin. Scale bar, 10 µm. **f** Quantification of F-actin intensities in lamellipodia of wild-type and mutant cells after subtraction of background from three independent experiments, B16-F1: $n = 85$ cells, mutant #10: $n = 86$ cells ($P$ value: 0.061 to B16-F1), mutant #33: $n = 86$ cells ($P$ value: 0.005 to B16-F1 and 1 to mutant #10). **g** Quantification of lamellipodia width in wild-type and mutant cells from three independent experiments, B16-F1: $n = 85$ cells, mutant #10: $n = 86$ cells ($P$ value: <0.001 to B16-F1), mutant #33: $n = 86$ cells ($P$ value: <0.001 to B16-F1 and 0.061 to mutant #10). **h** Loss of EFhD2 does not impair microspike formation. Quantification of microspikes in wild-type and mutant cells from three independent experiments, B16-F1: $n = 85$ cells, mutant #10: $n = 86$ cells ($P$ value: 0.979 to B16-F1), mutant #33: $n = 86$ cells ($P$ value: 0.535 to B16-F1 and 0.654 to mutant #10). **i** Loss of EFhD2 diminishes the efficacy of lamellipodium protrusion. Kymographs of representative phase-contrast movies are shown. **j** Multiple examples of lamellipodium protrusion in B16-F1 versus EFhD2-KO cells. **k** Quantification of protrusion rates from three independent experiments, B16-F1: $n = 20$ cells, mutant #10: $n = 29$ cells ($P$ value: <0.001 to B16-F1), mutant #33: $n = 23$ cells ($P$ value: 0.535 to B16-F1 and 0.475 to mutant #10). **c, f–h, k** Boxes in box plots indicate 50% (25–75%) and whiskers (5–95%) of all measurements, with dashed black lines depicting the medians, arithmetic means are highlighted in red. Non-parametric, Kruskal–Wallis test with Dunn's Multiple Comparison test (**c, f, k**) or one-way ANOVA with Tukey Multiple Comparison test (**g, h**) were used to reveal statistically significant differences between datasets. $P$ value: >0.05 (n.s.) <0.001 (***). All images shown are representative of three independent experiments unless otherwise specified.

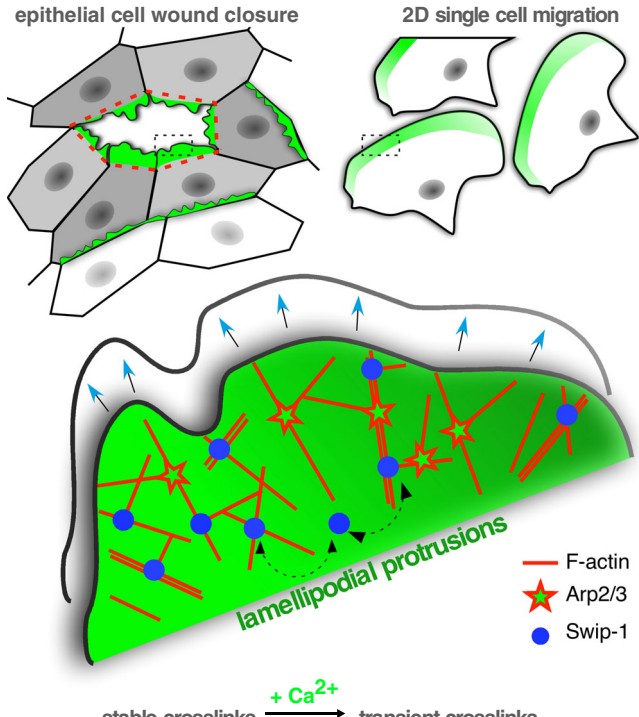

**Fig. 7 EFhD2/Swip-1 cross-linked lamellipodial actin networks drive single-cell migration and epithelial wound closure.** Schematic showing the proposed role of EFhD2/Swip-1 in regulating lamellipodial actin networks.

were obtained from the Bloomington stock center: w[1118] (BL3605), hmlΔ-Gal4 (BL30139), y[*] w[*]; P{w[+mC] = UAS-2xEGFP}AH3 (BL6658), w[*]; P{w[+mC] = UAS-Lifeact.GFP.W}3 (BL57326), y[1] w[*]; P{y[+t*] w[+mC] = UAS-Lifeact-Ruby}VIE-19A (BL35545), w[*]; PBac{w[+mC] = 20XUAS-IVS-NES-jRCaMP1a-p10}VK00005 (BL63792). RNAi against EFhD2/Swip-1: w[1118]; P{GD7047}v31307 was obtained from the Vienna Drosophila RNAi Center. Transgenic UAST-Swip, UAST-Swip-eGFP, UAST-eGFP-SwipΔEF, UAST-SwipΔCC-eGFP and UAST-SwipD82A/D118A-eGFP flies were generated using ΦC31- mediated transgenesis (y[1] M{vas-int.Dm}ZH2A w[*]; M{3xP3-RFP.attP'}ZH-86Fb (BL24749) and y[1] M{vas-int.Dm}ZH2A w[*]; M{3xP3-RFP.attP'}ZH-68E (BL24485)[70]. The *swip*[Δ1] mutant was generated by CRISPR/Cas9 of the following target sequence: 5'-GGGGTCTTCGAGAAGACCT-3'. Loss of the EFhD2/

Swip-1 protein was confirmed by Western blot analysis stained against His-Swip-1 (Pineda, Berlin).

**Protein purification.** Expression of GST-tagged human EFhD2, wild type, and mutant Swip-1 in *Escherichia coli* strain Rossetta 2 (Novagen) was induced with 0.75 mM IPTG at 21 °C for 16 h. The bacteria were harvested and lysed by ultra-sonication in PBS, pH 7.4 containing 5 mM benzamidine, 1 mM DTT, 5% (v/v) glycerol, 0.1 mM AEBSF and 2 units/mL Benzonase (Novagen). The fusion proteins were purified from supernatants of bacterial extracts by affinity chromatography using glutathione-conjugated agarose 4B (Macherey-Nagel). The GST-tag was subsequently cleaved off by PreScission protease (GE Healthcare) and the GST tag was absorbed on fresh glutathione-conjugated agarose. Swip-1 in the flow through was further purified by size-exclusion chromatography using a preparative HiLoad 26/75 Superdex column controlled by an Äkta Purifier System (GE Healthcare). Fractions containing Swip-1 were pooled, dialyzed against 30 mM Hepes, pH 7.4 containing 150 mM NaCl, 1 mM DTT, 5% (vol/vol) glycerol, snap-frozen in liquid nitrogen, and stored at −80 °C. Actin was extracted and purified from acetone powder of rabbit skeletal muscle using standard procedures[71]. Fractions were labeled on Cys374 with ATTO488 maleimide (ATTO-TEC) and stored in G-Buffer (5 mM Tris/HCl pH 8, 0.2 mM ATP, 0.5 mM DTT, 0.2 mM CaCl$_2$, 0.1 mg/mL NaN$_3$).

**Antibodies.** The rabbit anti-Swip-1 antibody was generated against the full-length *Drosophila* Swip-1 fused to a 6 × His-tag (pDEST17, ThermoFisher Scientific). The 6 × His-Swip fusion protein was expressed in *E. coli* and purified with Ni-NTA resin (GE Healthcare). Rabbits were immunized with purified proteins by Pineda Antikörper-service (Berlin, Germany).

For immunofluorescence of ex vivo *Drosophila* macrophages, the Swip (1:50,000 dilution) or ß-integrin (1:10 dilution, CF.6G11 from DSHB) antibodies were used. Primary antibodies were visualized with polyclonal Alexa Fluor-568-conjugated goat-anti-rabbit (1:1000 dilution; #A11036, Invitrogen) or Alexa Fluor-647-conjugated goat-anti-mouse (1:1000 dilution; #A21236, Invitrogen) antibodies, respectively. F-actin was visualized using Alexa Fluor-488 or Alexa Fluor-568-conjugated Phalloidin (1:100 dilution, #A12379 #A12380, Invitrogen) and nucleus by DAPI staining (1 µg/mL, #62248, Thermo Scientific). For immunofluorescence (B16-F1 cells) of primary polyclonal rabbit antibodies against VASP (1:1000 dilution[7]), cortactin (1:1000[7]), and WAVE2 (1:1000[7]) were used. Primary antibodies were visualized with polyclonal Alexa-555-conjugated (1:1000 dilution; #A21429, Invitrogen) or Alexa-488-conjugated (1:1000 dilution; #A-11034, Invitrogen) goat-anti-rabbit antibodies. The EGFP-signal of transfected cells expressing EGFP-tagged EFhD2/Swip-1 was enhanced with Alexa-488-conjugated nanobodies[72]. ATTO550-phalloidin (1:250 dilution, #AD550–82, Atto-Tec) was used for visualization of F-actin. For immunoblotting polyclonal rabbit anti-EFhD2/Swip-1 (from collaborator) antibody or mouse monoclonal antibody against glyceraldehyde-3-phosphate dehydrogenase (GAPDH) (1:1000; #CB1001-500UG, Merck) were used and visualized using phosphatase-coupled anti-rabbit (1:1000 dilution; #115-055-144, Dianova) or anti-mouse antibodies (1:1000 dilution; #115-055-62, Dianova). Anti-EFhD2 antibody was kindly provided by Dirk Mielenz, Erlangen.

**Validation of knock-out and knock-down of EFhD2/Swip-1.** For cell-type-specific knock-down in macrophages, EFhD2/Swip-1 RNAi lines were crossed with macrophage-specific hmlΔ driver line. Macrophages of 35 L3 larvae were isolated in 1 mL unsupplemented Schneider's medium, centrifuged for 5 min at 800 × g, and the supernatant carefully removed. Cells were resuspended in 10 μL 4 × SDS sample buffer and incubated at 95 °C for 10 min for SDS-PAGE. EFhD2/Swip-1 null-mutants were validated by decapitating 10 flies and squashing the heads in 30 μL 2 × SDS sample buffer and incubation at 95 °C for 10 min for SDS-PAGE. The following antibodies were used for Western Blot analysis: anti- EFhD2/Swip-1 (1:5000 dilution), Anti-Actin AB-5 (1:5000 dilution, BD Biosciences), IRDye 800CW Goat anti-Rabbit IgG and IRDye 680LT Goat anti-Mouse IgG Secondary Antibody (1:10,000 dilution, LI-COR).

**Cell culture and cell transfection.** *Drosophila* S2R+cells were propagated as described previously[73]. In short, cells were propagated in 1× Schneider's *Drosophila* medium (Gibco) supplemented with 10% FBS, 50 units/mL penicillin, and 50 μg/mL streptomycin in 25 cm² Corning cell culture flask (Thermo Fisher). S2R+cells were cotransfected with 0.6 μg Act5c-GAL4 and 1 μg DNA of respective gene of interest with a UAS sequence[74]. For estimation of the EFhD2/Swip-1 content in S2 and S2R+cells, recombinant EFhD2/Swip-1 and cell lysates were immunoblotted and band intensities were compared using LabImage 4.2.1 (Kapelan Bio-Imaging). S2 and S2R+cells were also stained with Hoechst 33342 (Thermo Scientific) for 30 min and imaged before settling on the surface with a Leica TCS SP8 with an HC PL APO CS2 ×63/1.4 oil objective. Cell and nucleus sizes were calculated assuming a circular shape. Molarity of Swip-1 in the cytosol was calculated with the known concentration of recombinant Swip-1 and specific cell number in the cell lysates of the immune blots.

B16-F1 (ATTC, CRL-6323) and derived cells were cultured in high-glucose DMEM culture medium (Lonza) supplemented with 1% penicillin-streptomycin (Biowest), 10% FBS (Biowest), and 2 mM UltraGlutamine (Lonza) at 37 °C in a 5% CO₂ atmosphere. B16-F1 cells were transfected with 1 μg plasmid DNA using JetPRIME transfection reagent (PolyPlus) in 35 mm diameter wells (Sarstedt) following the manufacturer's protocol. For the expression of EGFP-tagged EFhD2/Swip-1 in B16-F1 cells, a cDNA fragment encoding full-length EFhD2 was amplified from pCMV-Sport6-EFhD2 using the primers 5′-GTGAGATCTATGGCCACGGACGAGCTGGCCAC-3′ and 5′-CACGTCGACCTACTTAAAGGTGGACTGCAGCTC-3′ and inserted into the Bgl2-SalI sites of plasmid EGFP-C1 (Clontech). The Fidelity of generated plasmids was confirmed by sequencing.

**CRISPR/Cas9-mediated genome editing.** CRISPR/Cas9 technology was used to inactivate the *EFhD2/Swip-1* gene in B16-F1 mouse melanoma cells[75]. DNA target sequences of exon 1 were pasted into the CRISPR/Cas9 target online predictor tool CCTop (https://cctop.cos.uni-heidelberg.de:8043/) to generate sgRNA of 20 nucleotides with high-efficiency scores and minimal off-target efficiency and to cover all possible splice variants of the gene. The derived targeting sequence 5′-CGGCGCGCGGACCTCAACCA-3′ and the corresponding reverse oligonucleotide were hybridized and inserted into BbsI site of plasmid pSpCas9(BB)-2A-Puro(PX459)V2.0 (Addgene plasmid ID: 62988)[76]. Validation of CRISPR construct sequences was performed using a 5′-GGACTATCATATGCTTACCG-3′ sequencing primer. 24 h after transfection with the CRISPR construct, the cells were selected in a culture medium containing 2.5 μg/mL puromycin for 4 days and then cultivated for 24 h in the absence of puromycin. For isolation of clonal knockout cell lines, single cells were seeded by visual inspection into 96-well microtiter plates and expanded in pre-conditioned culture medium. Clones were analyzed by the TIDE sequence trace decomposition web tool (https://tide.deskgen.com[77]) and confirmed by immunoblotting using specific antibodies.

**TIRF microscopy.** TIRFM assays were performed in TIRF buffer (20 mM imidazole pH 7.4, 1 mM MgCl₂, 50 mM KCl, 0.5 mM ATP, 20 mM β-mercaptoethanol, 2.5 mg/mL methylcellulose (4000 cP), 15 mM glucose, 100 μg/mL glucose oxidase, and 20 μg/mL catalase), either in the absence or presence of 1 mM EGTA. In reactions without EGTA, a final concentration of 1 mM CaCl₂ was used. Concentrations of 0.1 μM or 1 μM and of EFhD2/Swip-1 were used for the TIRF assays to investigate the concentration- and Ca²⁺-dependent cross-linking behavior of EFhD2/Swip-1. The reactions were initiated by the addition of G-actin (0.5, 0.75, or 1.5 μM final concentration, 10% ATTO488-labeled at Cys374) and the mixtures were subsequently flushed into mPEG-silan (Mr 2000) (Lysan Bio)-pre-coated flow chambers. Higher concentrations of G-actin were used for the TIRF assays in presence of Ca²⁺ to account for the slower actin assembly rates under these conditions[78]. Images were acquired with a Nikon Eclipse TI-E inverted microscope equipped with a TIRF Apo ×100 NA 1.49 oil immersion objective at 0.5-s intervals with exposure times of 70 ms by an Ixon3 897 EMCCD camera (Andor) for at least 15 min.

**High- and low-speed sedimentation assays.** For high-speed sedimentation assays, 8 μM G-actin was polymerized in the presence of EFhD2/Swip-1 at the concentrations indicated either in 1 × KMEI (10 mM imidazole pH 7.0, 50 mM KCl, 1 mM MgCl₂, 1 mM EGTA) or 1 × KMI (10 mM imidazole pH 7.0, 50 mM KCl, 1 mM MgCl₂) with 1 mM CaCl₂ for 2 h at 4 °C. Subsequently, the samples

were centrifuged at 200,000 × g at 4 °C for 1 h and the pellets were brought to the original volume in 1 × SDS sample buffer. To quantitate co-sedimentation of EFhD2/Swip-1 with F-actin, after SDS-PAGE and Coomassie Blue staining, the amount of the proteins in the pellet and supernatant fractions was determined densitometrically using the ImageJ software. Calculations of free and bound EFhD2/Swip-1 were determined by the ratio of band intensities in respective fractions. The dissociation constants ($K_D$) and the number of EFhD2/Swip-1-binding sites ($n$) on actin was obtained by non-linear, least-square fitting assuming a model of independent, identical binding sites with the SOLVER plug-in in Excel (Microsoft) using the following equation:

$$[Swip_{bound}] = \frac{(n \cdot [actin_{total}] + [Swip_{total}] + K_D)}{2}$$
$$- \sqrt{\left(\frac{n \cdot [actin_{total}] + [Swip_{total}] + K_D}{2}\right)^2 - n \cdot [actin_{total}] \cdot [Swip_{total}]} \quad (1)$$

For low-speed sedimentation assays, 5 μM G-actin was polymerized in the presence of various concentrations of EFhD2/Swip-1 in the same buffers and at the same conditions as described above. The samples were then centrifuged at 20,000 × g at 4 °C for 1 h and the pellets were brought to the original volume in a 1×SDS sample buffer. To quantitate the extent of cross-linking induced by EFhD2/Swip-1, after SDS-PAGE and Coomassie Blue staining, the amount of F-actin in the pellet and supernatant fractions was determined densitometrically using Image J software.

**Immunohistochemistry and fluorescence staining.** Pupal macrophages were isolated as described previously[79]. In short, white to light brownish prepupae were collected and washed in 1 × PBS. The prepupae were moved to 1× Schneider's *Drosophila* medium (Gibco) supplemented with 10% FBS, 50 units/mL penicillin, and 50 μg/mL streptomycin and opened to rinse out the hemolymph. Cells were spread on ConcanavalinA (0.5 mg/mL, Sigma) coated glass coverslips for 1 h at 25 °C and subsequently fixed for 15 min with 4% paraformaldehyde in 1 × PBS. Cells were washed once with 1 × PBS+0.1% TritonX-100 and three times with 1 × PBS. If no antibody staining was performed, the washing step with PBS-T was omitted. Cells were stained with primary antibody overnight at 4 °C and secondary antibody with Phalloidin and DAPI for 45 min at room temperature in a humidified chamber. Stained cells were mounted in Mowiol 4–88 (Carl Roth). Cell size and morphology were analyzed by using Image J shape descriptors. Circularity ranges from 0 (infinitely elongated polygon) to 1 (perfect circle) (ImageJ, NIH). The length of randomly chosen focal adhesions in the lamellipodium of macrophages was measured using Image J.

The B16-F1 cells were fixed in pre-warmed PBS, pH 7.4 containing 4% paraformaldehyde and 0.06% picric acid for 20 min and subsequently washed three times with 100 mM glycine in PBS to quench the fixative. The cells were then permeabilized with 0.1% Triton X-100 in PBS for 30 s and blocked with PBG (PBS containing 0.045% cold fish gelatin (Sigma) and 0.5% BSA) for 30 min. Fixed specimens were incubated with primary antibodies overnight followed by incubation with secondary polyclonal goat-anti-rabbit antibodies for 2 h. ATTO550-phalloidin was used for the visualization of F-actin.

**Fluorescence microscopy.** Confocal images were taken with a Leica TCS SP8 with an HC PL APO CS2 ×63/1.4 oil objective and Leica Application Suite X (LasX) software. Structure illumination microscopic images were taken with a Zeiss ELYRA PS1 Microscope with a ×63/1.4 oil objective. Live imaging of macrophages was performed using a Zeiss CellObserver Z.1 with a Yokogawa CSU-X1 spinning disk scanning unit and an Axiocam MRm CCD camera (6.45 μm × 6.45 μm) and ZenBlue 2.5 software. Laser ablation of single cells was done using the UV laser ablation system DL-355/14 from Rapp OptoElectronics. Imaging of fixed B16-F1 cells was performed with an Olympus XI-81 inverted microscope equipped with an UPlan FI ×100/1.30NA oil immersion objective or a Zeiss LSM980 confocal microscope equipped with a Plan-Neofluar ×63/1.45NA oil immersion objective using 488 nm and 561 nm laser lines. Fluorescence intensities of phalloidin-stained lamellipodia were quantified from 8-bit images captured at identical settings using ImageJ software after background subtraction. Relative mean pixel intensities in lamellipodial regions of interest are shown as whiskers-box plots including all data points. The numbers of microspikes were manually counted.

**Live-cell imaging of pupal macrophages.** Live imaging of macrophages in pre-pupae was performed as previously reported[79]. In short, white prepupae were collected and glued to a glass coverslip on their dorsal-lateral side. Spinning disk time-lapse movies were taken with images every 20 s for 30 min. For tracking, a maximum projection of the acquired z-stack was obtained. Migrating macrophages were automatically tracked and manually corrected using the Imaris 9.3 software (Bitplane). For automatic detection of the cells, estimated diameter of 5 μm was assumed. Tracks were followed using an autoregressive motion algorithm to model the motion allowing a maximum distance of 10 μm between spots and a maximum gap size of 3. With manual correction of the tracks, gap sizes were reduced to a minimum. Specific values of Track Speed Mean, Track Displacement Length, and Track Straightness were obtained from the software.

**Live-cell imaging of wound closure of pupal epithelium**. Single epithelial cells of 18–20 h APF pupal abdomen were ablated using the UV laser ablation system DL-355/14 from Rapp OptoElectronics and wound closure was observed for 1 h every 30 s on a Zeiss CellObserver Z.1 with a Yokogawa CSU-X1 spinning disk scanning unit and an Axiocam MRm CCD camera (6.45 μm × 6.45 μm). Wound closure was analyzed by measuring the wound size in 5 min increments using freehand selections in Image J.

**Live-cell imaging of B16-F1 cells**. B16-F1 cells and EFhD2/Swip-1 KO mutant cells were seeded in low density onto 35 mm glass-bottom dishes (Ibidi) coated with 25 μg/mL laminin and allowed to spread for 3 h. To compensate for the lack of $CO_2$ during imaging, 25 mM HEPES, pH 7.0 was added to the medium. After mounting the chamber into the heating system (Ibidi), the cells were recorded by time-lapse phase-contrast imaging at 60 s intervals for 3 h using Olympus XI-81 inverted microscope (Olympus) driven by Metamorph software (Molecular Devices) and equipped with an Uplan FL N ×4/0.13NA objective (Olympus) and a CoolSnap EZ camera (Photometrics). Cells were tracked individually with the ImageJ Plugin MTrackJ. Cells that contacted each other or divided were excluded from analyses. Mean square displacements and cell speed were calculated in Excel (Microsoft) using a customized macro[80]. Lamellipodium protrusion was determined based on kymographs generated from time-lapse movies recording advancing lamellipodia at 5 s intervals over a time period of at least 10 min using a UPlan FI ×100/1.30NA oil immersion objective (Olympus). Kymographs were generated using ImageJ software by drawing lines from inside the cell and across the lamellipodium. Protrusion rates were calculated from respective slopes in kymographs.

**Statistical analysis**. Quantitative experiments were performed at least in triplicates to avoid any possible bias by environmental effects or unintentional error. Raw data were processed in Excel (Microsoft). Statistical analyses were performed using GraphPad Prism 7 (GraphPad) or Origin 2021 Pro (OriginLab). All data sets were tested for normality by the Shapiro–Wilk test. Statistical differences between normally distributed datasets of two groups were revealed by the $t$ test and not normally distributed datasets of two groups by the non-parametric Mann–Whitney $U$ rank-sum test. The Mann–Whitney test was used and $P$ value (two-tailed) was obtained ($P$ value: 0.12 (ns), 0.033 (*), 0.002 (**), <0.001 (***)). For comparison of more than two groups, the statistical significance of normally distributed data was examined by one-way ANOVA followed by Tukey Multiple Comparison. In the case of not normally distributed data, the non-parametric Kruskal–Wallis test followed by Dunn's Multiple Comparison was used. Statistical differences are reported as *$p < 0.05$, **$p < 0.01$, ***$p < 0.001$, and n.s.

**Reporting summary**. Further information on research design is available in the Nature Research Reporting Summary linked to this article.

## Data availability

The data that support the findings of this study are available within the article, Supplementary Information, or from the corresponding author upon reasonable request. The raw gene expression data generated in this study have been deposited in the NCBI Gene Expression Omnibus (GEO) database under accession code GSE200966. Source Data are provided with this paper.

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

## Acknowledgements

We thank the Bloomington Stock Center and VDRC for fly stocks, K. Ramlow for purifying and testing *Drosophila* anti-Swip-1 antibodies, A. Hirschhäuser for isolating the macrophages for bulk RNA seq analysis, T. Kaufmann for support in interpretation of structural data, and D. Mielenz for providing the rabbit anti-mouse EFhD2/Swip-1 antibody and human EFhD2 cDNA. The work was supported by grants to S.B. (BO1890/4-1) and J.F. (Fa330/9-2) from the Deutsche Forschungsgemeinschaft (DFG) and by J.G. from the VW Stiftung, "Big Data in den Lebenswissenschaften der Zukunft", Nr. A129197.

## Author contributions

S.B. and J.F. designed the project, made the figures, and wrote the manuscript. F.L. performed all Drosophila experiments. T.P. performed all B16-F1 mouse melanoma work and biochemical in vitro experiments. S.P. performed the bulk RNA seq analysis. G.S. and J.G. managed and coordinated RNA seq analysis. K.R. performed bioinformatical analyses. All authors commented on the manuscript.

## Funding

## Competing interests

The authors declare no competing interests.
