## [Peer Review File · Nature Communications]

Calcium bursts allow rapid reorganization of EFhD2/Swip-1 crosslinked actin networks in epithelial wound closureReviewers' Comments:

Reviewer #1:

Remarks to the Author:

The authors have studied the function of EFHD2/Swip-1 in activated *Drosophila* macrophages. The authors find that EFHD2/Swip-1 reorganizes the lamellipodia upon calcium stimulus and that EFHD2/Swip-1 acts as an actin-crosslinking protein rather than an actin-bundling protein. They also show that EFHD2/Swip-1 is recruited to a single cell wound upon calcium bursts. Overall, this work provides a better understanding of how EFHD2/Swip-1 is regulated by calcium in order to modify the actin architecture at the lamellipodia and promote cell migration.

An important point to modify is that the authors make some strong claims in their abstract/introduction/results of having identified a novel role for EFHD2/Swip-1 in lamellipodia formation and cell migration. However, these cellular functions have already been published. Swip1 has been shown to localize to lamellipodia and to positively regulate cell migration both via its lamellipodial function as well as through facilitating integrin traffic and adhesion turnover in cancer cells (Huh et al., *Cell Mol Life Sci* 2013, Huh et al., *Oncotarget* 2015, Moreno-Layseca et al., *Nat Cell Biol* 2021). Conversely, Swip1 has been shown to negatively regulate cell migration in human B-cells (Reimer et al., 2020 *Cell Reports*). Some of these previous studies are mentioned in the discussion. However, they should be mentioned in the introduction and in the results in the context of the authors' data where they find the same: EFHD2/Swip-1 reducing migration in immune cells in vivo (Figure 3) and increasing migration in murine melanoma cells in vitro (Figure 4). The authors should modify their manuscript and the way these data are described, cite these previous publication in their intro/results and focus on highlighting the main novel finding (a calcium-regulated actin-crosslinking function for EFHD2) and describe their migration regulating experiments as confirmatory data reproducing the pro-migratory role of Swip1 in another system. These experiments are supportive of their main novel finding. The main novel finding is interesting but somewhat preliminary at the moment. The authors should identify and generate a calcium-binding deficient mutant of Swip-1 (possibly taking advantage of the recent crystal structure analysis of the calcium-bound EFHD2 core domain of EFHD2/Swip-1) and demonstrate mechanistically what is the link between Swip-1 calcium-binding, its actin cross-linking activity and the cell phenotypes (migration/wound healing). Ideally, these studies should be supported by experiments with actinin binding deficient Swip-1 mutants that have been described in the literature.

Other points:

Line 50. EFHD2 has already been identified as a pro-migratory protein at the lamellipodia in mammalian macrophages and melanoma cells (Tu et al., *Int Immunopharmacol* 2018, Huh et al., *Oncotarget* 2015; Kwon et al., *Plos One* 2013). Please rephrase or remove this sentence.

Line 52. The authors mention that EFHD2/Swip-1 is upregulated in activated *Drosophila* macrophages, but they do not mention compared to which condition (quiescent larval macrophages?).

Line 53. Again, the role of EFHD2 in cell migration has been already reported (Tu et al., *Int Immunopharmacol* 2018, Huh et al., *Oncotarget* 2015; Kwon et al., *Plos One* 2013). Please rephrase or remove this sentence.

Line 122. The authors mention the presence of intracellular vesicles in GFP-EFHD2-expressing cells in Figure 4a. Thus, the recently identified role for EFHD2 in endosomes should also be mentioned in conjunction to this observation (Moreno-Layseca et al., *NCB* 2021).

Figure 1. In contrast to EFHD2, EFHD1 only contains one EF-hand domain. Could the authors clarify this in panel C? Right now, this panel shows EFHD1 as if it included two EF-hand domains.

There are some mistakes in the figure numbering between the text and the figures: text lines 180-190 refers to Figure 1g, 1e and 1f. I believe these should be 2g, 2e, 2f.

Line 186. Since the authors use both mammalian and *Drosophila*-generated cell lines, they should mention the origin of S2R+ cells for clarity.

Line 241. The effect of EFHD2/Swip-1 on melanoma cells has already been reported. This needs to be acknowledged with the appropriate reference (Huh et al., *Oncotarget* 2015).

Line 246. The opposite results in cell migration upon EFHD2/Swip-1 loss in melanoma and *Drosophila*

macrophages are intriguing and in line with previous studies in human cancer cell lines and B-cells. Do the authors have any ideas to what could the difference be attributed? Different modes of migration (amoeboid/adhesion independent vs. mesenchymal/adhesion dependent)? Cell-type specific functions for EFHD2: immune cells vs. other cell types?

Line 266. The authors should attempt to compare the cytoplasmic concentrations of EFHD2/Swip-1 (or at least the expression levels with a western blot) in the cell lines used in the manuscript in order to relate the functions observed, for example, in the migratory effect observed in melanoma cells vs. *Drosophila* macrophages.

Line 405. The authors need to be more cautious when discussing migration in 2D vs. 3D. Although it is true that migration mode-specific roles have been suggested for EFHD2/Swip-1 (Moreno-Layseca et al., NCB 2021), all the studies referred by the authors in this paragraph have looked at migration of immune cells and cancer cells both in 2D surfaces, and found the same effect of EFHD2/Swip-1 on cell migration. Moreover, some of these studies also utilize different 3D and in vivo migration and invasion models and still found the same effect as they did on 2D surfaces upon EFHD2/Swip-1 depletion or overexpression. Thus, such a general conclusion does not apply in this case and a different explanation should be considered for the opposite effects observed in the migration of *Drosophila* macrophages.

Line 514. The authors need to specify the species of the target protein used to generate the rabbit anti-Swip-1 antibody: full length Swip-1 from *Drosophila*?

Minor points:

Line 139 and 142. Please consider rephrasing to avoid the same beginning of the sentences "Previous studies".

Line 179. Typo: "in a punctate pattern"

Line 527. Please specify who is the collaborator who provided the rabbit anti-EFHD2/Swip-1 antibody. Dirk Mielenz is mentioned later (line 531), is this the same collaborator/antibody?

Line 665. Please specify the parameters used in the IMARIS algorithm to calculate the speed (for example, maximum gap size between spots, algorithm used to model the movement: Brownian movement?)

Reviewer #2:

Remarks to the Author:

Lehne et al.

This manuscript addresses an important question regarding the role of the actin-binding protein Swip-1 in lamellipodial protrusions and cell migration. It is well established that lamellipodia form at the leading edge of a migrating cell and drives migration. The branched network of actin filaments nucleated by the Arp2/3 complex provides the mechanical forces for movement. Much is known about the branched actin cytoskeletal network of lamellipodia, but many questions remain, in particular, how the actin-binding protein Swip-1 regulates protrusive forces generated by the actin cytoskeleton in a Ca⁺⁺-dependent manner.

To answer these questions, the authors used *drosophila* hemocytes to identify actin binding proteins that are upregulated in activated macrophages. RNA seq data identified over 800 genes upregulated in prepupal macrophages. They narrowed the list to 50 cytoskeletal/motility genes upregulated in prepupae and focused attention on Swip-1, known as a Ca⁺⁺-dependent actin bundling protein.

The data showed that Swip-1, an EF-hand containing protein, is upregulated in prepupal macrophages by 2.46-fold and localizes to the lamellipodia. Loss of function affects cell migration in fly macrophages and mouse melanoma cells although the effects on migration were dissimilar. *Drosophila* macrophages showed an increase in the speed of migration while mouse melanoma cells showed a decrease in

speed of migration. In vitro studies showed that Swip-1 does not bundle but cross-links actin filaments in a Ca^{++} -dependent manner.

Using a single-cell wounding assay, the authors showed that cells surrounding the wound site elaborated broad lamellipodia, the formation of which was accompanied by an increase in Ca^{++} . A slow Ca^{++} wave beginning in the cells surrounding the wound site spread outward to neighboring rows of cells. The authors concluded that the transient increase in $[Ca^{++}]$ reduces Swip-1 cross-linking activity and thereby promote rapid reorganization of the actin filament network to drive epithelial wound closure.

- What are the noteworthy results?

The novel findings are that Swip-1 cross-links rather than bundles actin filaments in a Ca^{++} -dependent manner. Deletion of Swip-1 alters cell migration. The effects on migration differ in different cell types.

- Will the work be of significance to the field and related fields? How does it compare to the established literature? If the work is not original, please provide relevant references.

The work documents the function of Swip-1 both in vivo and in vitro. The wound healing assay supports a role for Swip-1 in lamellipodial protrusion accompanied by a transient increase in Ca^{++} .

- Does the work support the conclusions and claims, or is additional evidence needed?

- Are there any flaws in the data analysis, interpretation and conclusions? - Do these prohibit publication or require revision?

The work supports the conclusions, but questions remain about whether the activity of Swip-1 functions as a brake of an accelerator of cell migration.

The authors tracked the movement of cells to determine the average speed of migration as a measure of Swip-1 role in cell migration. Unfortunately, the average speed of migration is not sufficient to capture the complexity of cell migration in different cell types and different environments. A more substantive analysis of the motility data including instantaneous velocity, pathlength and directionality might give clues to the difference observed between the 2 cell types.

- Is the methodology sound? Does the work meet the expected standards in your field?

- Is there enough detail provided in the methods for the work to be reproduced?

The methodology is sound, and the details provided as sufficient to replicate the experiments.

Specific changes in the manuscript.

1. line 94 "shorter than implied"

2. line 127 "results explain"

3. line 184 Figure 2e, e'

4. line 186 Figure 2f

5. line 190 Figure 2g

6. line 321, the term leading edge is used incorrectly. Cells have leading edges, not the wound.

Suggested wording "actin assembled to form broad lamellipodial protrusions in cells surrounding the wound site"

7. line 326 "edge of cells surrounding the wound"

8. line 330 ditto

9. line 332 "cells several rows back from the wound site exhibited a delayed response"

10. line 347, awkward phrasing, clarify the direction of the change in $[Ca^{++}]$

References to figures 5 and 6 are misquoted in the text.

Reviewer #3:

Remarks to the Author:

The manuscript from Lehne et al describes a study centering on the sole *Drosophila* homolog of EFhD2/Swip-1 proteins, calcium-sensitive adapters that operate in diverse tissue and cell-type setting, but whose specific molecular functions, particularly vis-à-vis the actin-based cytoskeleton, have remained somewhat ambiguous. The study encompasses a rather broad range of topics and employs a variety of experimental approaches and techniques, which are carried out in competent fashion.

I wish to state at the outset of this review that, in my judgement, a primary drawback of the manuscript is a lack of focus, making it difficult for the reader to assess the novelty and significance of the reported findings. Two major, related advances are put forward (Figures 5-7 and related text): *Drosophila* EFhD2/SWIP-1 is shown to act as a microfilament cross-linker (rather than as a microfilament bundling protein), a capacity that is inhibited by calcium: EFhD2/SWIP-1 function is necessary for proper repair/closure of epithelial wounds in *Drosophila* pupae, a process involving calcium-regulated activity of the actin-based cytoskeleton. However, the first half or so of the manuscript is devoted to presentation of data which, to a large part, have been reported previously or are only marginally relevant to the key observations described and needlessly complicate matters (see below). I thus strongly recommend that the authors substantially re-organize and shorten the manuscript, as well as take into account specific additional issues detailed below.

1. An aspect of EFhD2/SWIP-1 localization (via antibodies) that should be clarified (Figure 2 and accompanying text): Besides the general localization to the broad lamellipodia of pupal macrophages, EFhD2/SWIP-1 corresponds quite clearly - together with F-actin- to radial structures. I don't believe that these are mentioned in the text. Can the authors add comments on these structures? Are they perhaps associated with focal adhesions? Why are they observed only in the F-actin panel when visualized using super-resolution microscopy?

2. Figures 3 and 4 and the accompanying text describe experiments primarily designed to assess the functional role(s) of *Drosophila* EFhD2/SWIP-1 in pupal macrophages and of mouse EFhD2/SWIP-1 in a melanoma cell line. In keeping with the comments above, I found much of these data to be problematic on various levels. The reported knockout phenotypes, even if statistically significant, are not particularly compelling, do not provide true insight, and are in some cases perplexing and difficult to explain (eg- the opposite effects on motility in the fly and murine systems). Furthermore, previous publications (from the GIST (Gwangju) groups and others) have addressed much of what is reported here regarding mammalian EFhD2/SWIP-1, even if somewhat less rigorously.

3. A major claim made here is that EFhD2/SWIP-1 acts to cross-link rather than bundle microfilaments, and does so in a calcium-dependent manner. The key experiment (Figure 5h and movie M4) involves visualization of microfilament network morphologies using TIRF microscopy, in which *Drosophila* EFhD2/SWIP-1 function is compared with two established (human) actin bundling proteins, fascin and alpha actinin 4 (this established characteristic of fascin and actinin 4 should be better emphasized in the text). Given that these novel findings contrast with previous studies of human EFhD2/SWIP-1, as acknowledged by the authors, the human homolog should be subjected to this assay as well.

4. Returning now to the earlier remarks regarding manuscript organization. I believe that the manuscript will be vastly improved by revising and shortening the main text and associated data, so as to focus primarily on what now constitutes the second half of the paper (Figs 5-8). The material that currently appears earlier is, to my mind, inconclusive and distracting, and should be used sparingly, primarily to introduce EFhD2/SWIP-1 proteins and the *Drosophila* homolog. As noted, the experiments involving the mouse cell line are particularly problematic and, in my judgement, should not be included at all. Re-organizing the manuscript in this fashion is necessary to allow readers to recognize the novel observations reported and properly appreciate their significance.

Minor issues

Figure 2e- what are the arrows pointing to? Also- the related text (line 184) erroneously refers to Figure 1 rather than Figure 2.

Figure 2f- Swip-1-eEFP should be changed to Swip-1-eEGP.

Lines 338-339- The authors reference two previous *Drosophila* studies that described calcium influxes in the context of wound healing, stating that both involved the embryonic epidermis. While this is true of Razzell et al, the Antunes et al study (reference #48) focused on the pupal notum.

Figure 6d- what are the white asterisks marking?

REVIEWERS' COMMENTS

Reviewer #1 (Remarks to the Author):

The authors have studied the function of EFHD2/Swip-1 in activated *Drosophila* macrophages. The authors find that EFHD2/Swip-1 reorganizes the lamellipodia upon calcium stimulus and that EFHD2/Swip-1 acts as an actin-crosslinking protein rather than an actin-bundling protein. They also show that EFHD2/Swip-1 is recruited to a single cell wound upon calcium bursts. Overall, this work provides a better understanding of how EFHD2/Swip-1 is regulated by calcium in order to modify the actin architecture at the lamellipodia and promote cell migration.

We thank the reviewer for pointing this out.

An important point to modify is that the authors make some strong claims in their abstract/introduction/results of having identified a novel role for EFHD2/Swip-1 in lamellipodia formation and cell migration. However, these cellular functions have already been published. Swip1 has been shown to localize to lamellipodia and to positively regulate cell migration both via its lamellipodial function as well as through facilitating integrin traffic and adhesion turnover in cancer cells (Huh et al., *Cell Mol Life Sci* 2013, Huh et al., *Oncotarget* 2015, Moreno-Layseca et al., *Nat Cell Biol* 2021).

We agree with the referee and restructured our manuscript, particularly we now focused on presenting our data in a more straightforward manner within the context of what has been previously published. Notably, we considered previous publications in the introduction and in the results in the context of our data, in particular from the GIST (Gwangju) groups previously describing a role of EFHD2/Swip-1 in lamellipodia formation and cell migration in HeLa cells and B16-F10 mouse melanoma cells (Kwon et al., 2013, *PlosOne*; Huh et al., *Cell Mol Life Sci* 2013, Huh et al., *Oncotarget* 2015). Without any intention to diminish previous work, we feel that our rigorous single-cell analysis of B16-F1 mouse melanoma cells goes beyond this point and provides, for the first time, a molecular explanation for the reduced protrusion and migration rates in genetic knockout mutant cells lacking EFHD2. Combined, these data strongly suggest that cross-linking of adjacent actin filaments within

dendritic arrays by EFHD2 significantly contributes to mechanical stability of lamellipodia to drive efficient protrusion.

Conversely, Swip1 has been shown to negatively regulate cell migration in human B-cells (Reimer et al., 2020 Cell Reports). Some of these previous studies are mentioned in the discussion. However, they should be mentioned in the introduction and in the results in the context of the authors' data where they find the same: EFHD2/Swip-1 reducing migration in immune cells *in vivo* (Figure 3) and increasing migration in murine melanoma cells *in vitro* (Figure 4). The authors should modify their manuscript and the way these data are described, cite these previous publication in their intro/results and focus on highlighting the main novel finding (a calcium-regulated actin-crosslinking function for EFHD2) and describe their migration regulating experiments as confirmatory data reproducing the pro-migratory role of Swip1 in another system.

As suggested by the reviewer, we now refer better to previous studies on EFHD2/Swip-1 function in immune cells published by the Mielenz group (Reimer et al., 2020 Cell Reports), equally in the introduction, results and in the discussion. More importantly, we now provide additional evidence for a conserved function of EFHD2/Swip-1 in both regulating lamellipodial actin networks and integrin adhesion dynamics. In detail, we found that isolated *swip-1* mutant macrophages exhibited increased β -integrin-marked focal adhesion sites resulting in an increased cell spreading (cell size) when plated on surfaces coated with the ECM protein vitronectin (see new supplementary figure S2). Thus, enlarged focal adhesions caused by slower integrin turnover might further contribute to an increase in the migration speed as similarly observed for *zyxin*-deficient macrophages in *Drosophila* (Moreira et al., 2013). Such a prominent inverse relationship between focal adhesion size and cell migration speed has been shown in numerous motile cell types (Kim et al., 2013) and it might be also relevant in B-cells (Reimer et al., 2020 Cell Reports). We discussed this as an additional mechanism explaining an increase of cell migration speed of mutant immune cells *in vivo* (see discussion).

These experiments are supportive of their main novel finding. The main novel finding is interesting but somewhat preliminary at the moment. The authors should identify and generate a calcium-binding deficient mutant of Swip-1 (possibly taking

advantage of the recent crystal structure analysis of the calcium-bound EFhD2 core domain of EFhD2/Swip-1) and demonstrate mechanistically what is the link between Swip-1 calcium-binding, its actin cross-linking activity and the cell phenotypes (migration/wound healing). Ideally, these studies should be supported by experiments with actinin binding deficient Swip-1 mutants that have been described in the literature.

This is an excellent idea. As suggested, we now analyzed the functional relevance of calcium binding of Swip-1 *in vivo* and *in vitro*. Based on sequence alignment, *Drosophila* Swip-1 has two predicted conserved EF-hand motifs, EF-loop 1, 82aa DTARDGFLDLQE 93aa (corresponding to human EFhD2 105-116aa), and EF-loop 2, 118aa DEDNDGKISFRE 129aa (corresponding to human EFhD2 141-152aa). Mutations in the first highly conserved aspartate residues of both EF-loops (D82A/D118A; see supplementary figure S1, marked by red boxes) have been shown to abolish calcium binding capacity (Ferrer-Acosta et al., 2013; Gutierrez-Ford et al., 2003). Interestingly, re-expression of the mutant Swip-1-D82A/D118A transgene failed to rescue migration defects of *swip-1* deficient macrophages (quantification in figure 2m). Remarkably, Swip-1-D82A/D118A mutant protein was no longer enriched in actively protruding lamellipodia, but rather re-localized to the cell cortex suggesting that calcium or full activity might be critical for lamellipodial localization of Swip-1 (new figure 2n; new supplementary movie M3). Moreover, mutant macrophages re-expressing Swip-1-D82A/D118A often attached together and formed clusters (Figure 2n). Cell cluster formation was also prominent in isolated mutant macrophages as well as mutant cells re-expressing Swip-1-D82A/D118A plated on surfaces coated with the ECM protein vitronectin. As described above, we found an increased β -integrin accumulation at cell-cell contacts (Supplementary figure 2d). Mutant cells showed an enhanced spreading and increased β -integrin-marked focal adhesion sites along the leading edge (see quantification in supplementary figure 2e, f) suggesting a possible conserved role of Swip-1 in regulating integrin adhesion dynamics as recently reported in human breast cancer cell line (Moreno-Layseca et al., 2021).

To further assess the functional relevance of calcium binding of Swip-1 on actin filament architecture *in vitro*, we also purified a recombinant calcium-binding deficient

Swip-1-D82A/D118A mutant. TIRF analysis revealed that this mutant protein still induced stable, perpendicular cross-links, albeit in a calcium-independent manner as expected. Unexpectedly, however, this mutant Swip-1-D82A/D118A protein additionally exhibited prominent actin-bundling activity.

Of note, the determined X-ray structure of human EFhD2 (Park et al., 2016) and the predicted 3D structure of *Drosophila* Swip-1 computed by AlphaFold (Jumper et al. 2021) could be precisely superimposed, thus excluding major conformational differences as the underlying reason explaining differential bundling or cross-linking of actin filament in the absence of calcium. On the other hand, conformational changes in EF hands triggered by calcium binding, as previously shown for human EFhD2 (Park et al., 2016), are likely to be responsible for the dissociation of cross-links formed by EFhD2 and Swip-1. We further hypothesize, that the acquired bundling activity of the Swip-1 mutant could also be driven by enhanced electrostatic interactions of the negative actin molecule with Swip-1, in which two negative residues were replaced by alanine (D82A/D118A). We included this in discussion.

Other points:

Line 50. EFHD2 has already been identified as a pro-migratory protein at the lamellipodia in mammalian macrophages and melanoma cells (Tu et al., Int Immunopharmacol 2018, Huh et al., Oncotarget 2015; Kwon et al., Plos One 2013). Please rephrase or remove this sentence.

We have removed this sentence.

Line 52. The authors mention that EFHD2/Swip-1 is upregulated in activated *Drosophila* macrophages, but they do not mention compared to which condition (quiescent larval macrophages?).

We apologize for the unclear description. We now included further information in the abstract as follows “...at the onset of metamorphosis when macrophage behavior shifts from quiescent to migratory state.”

Line 53. Again, the role of EFHD2 in cell migration has been already reported (Tu et

al., *Int Immunopharmacol* 2018, Huh et al., *Oncotarget* 2015; Kwon et al., *Plos One* 2013). Please rephrase or remove this sentence.

We have rephrased this sentence accordingly, and point out that “Loss- and gain-of-function analysis confirm a critical function of EFhD2/Swip-1 in lamellipodial cell migration in fly and mouse melanoma cells.”

Line 122. The authors mention the presence of intracellular vesicles in GFP-EFHD2-expressing cells in Figure 4a. Thus, the recently identified role for EFHD2 in endosomes should also be mentioned in conjunction to this observation (Moreno-Layseca et al., *NCB* 2021).

Thank you We have included the missing reference.

Figure 1. In contrast to EFHD2, EFHD1 only contains one EF-hand domain. Could the authors clarify this in panel C? Right now, this panel shows EFHD1 as if it included two EF-hand domains.

Both, EFhD1 and EFhD2 contain two calcium-binding EF-loops as confirmed by recent structure analysis (Mun et al., 2020). We have included this reference in the revised manuscript.

There are some mistakes in the figure numbering between the text and the figures: text lines 180-190 refers to Figure 1g. 1e and 1f. I believe these should be 2g, 2e, 2f.

We apologize for this mistake. Previous figure 1 is now supplementary figure S1, whereas previous figure 2 is now main figure 1.

Line 186. Since the authors use both mammalian and *Drosophila*-generated cell lines, they should mention the origin of S2R+ cells for clarity.

We apologize for the missing information. The macrophage-like *Drosophila* S2R+ cells have been isolated from original Schneider (S2) cells expressing the Wg

receptor, Dfrizzled-2 described by Yanagawa et al., 1998. We have included this important reference.

Line 241. The effect of EFHD2/Swip-1 on melanoma cells has already been reported. This needs to be acknowledged with the appropriate reference (Huh et al., Oncotarget 2015).

We have included the missing information.

Line 246. The opposite results in cell migration upon EFHD2/Swip-1 loss in melanoma and *Drosophila* macrophages are intriguing and in line with previous studies in human cancer cell lines and B-cells. Do the authors have any ideas to what could the difference be attributed? Different modes of migration (amoeboid/adhesion independent vs. mesenchymal/adhesion dependent)? Cell-type specific functions for EFHD2: immune cells vs. other cell types?

As discussed in more detail, this difference might be due to different protein requirements as well as different physical requirements for the distinct migration modes. The architecture of the actin cytoskeleton highly depends on substrate rigidity and cells adapt to different extracellular resistance by reorganizing their actin network (Lo et al., 2000; Mueller et al., 2017). B16-F1 cells migrate on a rigid 2D substrate coated mostly with laminin. Thus, without actin bundling and cross-linking activity of EFhD2, even in the presence of other cross-linkers such as α -actinin, the Arp2/3 complex might be insufficient for maintaining the mechanical stability of the branched actin network at the leading edge of B16-F1 cells. As a consequence, loss of EFhD2 in B16-F1 cells caused a significant reduction of protrusion rate, cell speed and effective migration. By contrast, immune cells migrate across a complex 3D environment with highly varying extracellular resistance that requires more dynamic protrusions. Thus, the loss of Swip-1 in *Drosophila* macrophages would result in a reduced actin network density that could lead to faster but less persistent migration *in vivo*. Supporting this notion, we found that forced overexpression of Swip-1 stabilizes protrusions at the expense of migration speed *in vivo*. Thus, the assembly of cross-linking proteins such as EFhD2/Swip-1 into the branched actin network might contribute to a resistance-adaptive behavior of migrating cells as recently proposed

(Chen et al., 2020). In addition, EFhD2/Swip-1 has been recently identified as a cargo-specific adaptor for CG-endocytosis controlling integrin adhesion turnover (Moreno-Layseca et al., 2021). In the revised manuscript we provide first evidence that this function in integrin adhesion is conserved between humans and flies, and might also contribute to changed migratory behavior of macrophages *in vivo*. In particular, we found that *swip-1* deficient macrophages showed an increase in integrin-marked adhesions marked by anti- β -integrin immunostainings. Enlarged focal adhesion caused by slower integrin turnover might further contribute to an increase in the migration speed as similarly observed for *zyxin*-deficient macrophages in *Drosophila* (Moreira et al., 2013). Such a prominent inverse relationship between focal adhesion size and cell migration speed has been shown in numerous motile cell types (Kim and Wirtz, 2013).

Line 266. The authors should attempt to compare the cytoplasmic concentrations of EFHD2/Swip-1 (or at least the expression levels with a western blot) in the cell lines used in the manuscript in order to relate the functions observed, for example, in the migratory effect observed in melanoma cells vs. *Drosophila* macrophages.

We fully agree that this information would have been helpful. In fact, we tried to determine the cellular concentration of EFhD2 in B16-F1 cells by titrating different amounts of recombinant human EFhD2 and cell lysates with a defined number of B16-F1 cells in immunoblots. However, due to the different affinities of our polyclonal antibody for human *versus* mouse EFhD2 (provided by Dirk Mielenz), this was unfortunately not possible. Moreover, alternative and commercially available antibodies that potentially react equally well with human or murine EFhD2 could not be delivered owing to the current pandemic-related supply shortages in due time. Thus, this important aspect has to be addressed in future work. In the current study, we therefore compared the activities of human EFhD2 in our TIRF experiments, analogous to *Drosophila* Swip-1, at concentrations of 0.1 and 1 μ M, respectively.

Line 405. The authors need to be more cautious when discussing migration in 2D vs. 3D. Although it is true that migration mode-specific roles have been suggested for EFHD2/Swip-1 (Moreno-Layseca et al., NCB 2021), all the studies referred by the

authors in this paragraph have looked at migration of immune cells and cancer cells both in 2D surfaces, and found the same effect of EFHD2/Swip-1 on cell migration. Moreover, some of these studies also utilize different 3D and in vivo migration and invasion models and still found the same effect as they did on 2D surfaces upon EFHD2/Swip-1 depletion or overexpression. Thus, such a general conclusion does not apply in this case and a different explanation should be considered for the opposite effects observed in the migration of *Drosophila* macrophages.

As mentioned above, we now provide new evidence for a conserved function of EFHD2/Swip-1 in both regulating lamellipodial actin networks and integrin adhesion dynamics of *Drosophila* macrophages. In detail, we found that isolated *swip-1* mutant macrophages exhibited increased β -integrin-marked focal adhesion sites resulting in an increased cell spreading (cell size), when plated on surfaces coated with the ECM protein vitronectin (see new supplementary figure S2). Thus, enlarged focal adhesions caused by slower integrin turnover might further contribute to an increase in the migration speed as similarly observed for *zyxin*-deficient macrophages in *Drosophila* (Moreira et al., 2013). Such a prominent inverse relationship between focal adhesion size and cell migration speed has been shown in numerous motile cell types (Kim et al., 2013) and it might be also relevant in B-cells (Reimer et al., 2020 Cell Reports). We discuss this as an additional mechanism explaining an increase of cell migration speed of mutant immune cells *in vivo* (see discussion).

Line 514. The authors need to specify the species of the target protein used to generate the rabbit anti-Swip-1 antibody: full length Swip-1 from *Drosophila*?

Thank you. We have included the missing information in the materials and methods section.

Minor points:

Line 139 and 142. Please consider rephrasing to avoid the same beginning of the sentences "Previous studies".

The text has been changed accordingly.

Line 179. Typo: “in a punctate pattern”

The text has been changed accordingly.

Line 527. Please specify who is the collaborator who provided the rabbit anti-EFHD2/Swip-1 antibody. Dirk Mielenz is mentioned later (line 531), is this the same collaborator/antibody?

Yes, the text has been changed accordingly.

Line 665. Please specify the parameters used in the IMARIS algorithm to calculate the speed (for example, maximum gap size between spots, algorithm used to model the movement: Brownian movement?)

For automatic detection of the cells an estimated diameter of 5 μ m was assumed. Tracks were followed using an autoregressive motion algorithm to model the motion allowing a maximum distance of 10 μ m between spots and a maximum gap size of 3 μ m. With manual correction of the tracks, gap sizes were reduced to a minimum. We included this information in the materials and methods section.

Reviewer #2 (Remarks to the Author):

Lehne et al.

This manuscript addresses an important question regarding the role of the actin-binding protein Swip-1 in lamellipodial protrusions and cell migration. It is well established that lamellipodia form at the leading edge of a migrating cell and drives migration. The branched network of actin filaments nucleated by the Arp2/3 complex provides the mechanical forces for movement. Much is known about the branched actin cytoskeletal network of lamellipodia, but many questions remain, in particular, how the actin-binding protein Swip-1 regulates protrusive forces generated by the actin cytoskeleton in a Ca⁺⁺-dependent manner. To answer these questions, the

authors used drosophila hemocytes to identify actin binding proteins that are upregulated in activated macrophages. RNA seq data identified over 800 genes upregulated in prepupal macrophages. They narrowed the list to 50 cytoskeletal/motility genes upregulated in prepupae and focused attention on Swip-1, known as a Ca^{++} -dependent actin bundling protein.

The data showed that Swip-1, an EF-hand containing protein, is upregulated in prepupal macrophages by 2.46-fold and localizes to the lamellipodia. Loss of function affects cell migration in fly macrophages and mouse melanoma cells although the effects on migration were dissimilar. Drosophila macrophages showed an increase in the speed of migration while mouse melanoma cells showed a decrease in speed of migration. In vitro studies showed that Swip-1 does not bundle but cross-links actin filaments in a Ca^{++} -dependent manner. Using a single-cell wounding assay, the authors showed that cells surrounding the wound site elaborated broad lamellipodia, the formation of which was accompanied by an increase in Ca^{++} . A slow Ca^{++} wave beginning in the cells surrounding the wound site spread outward to neighboring rows of cells. The authors concluded that the transient increase in $[Ca^{++}]$ reduces Swip-1 cross-linking activity and thereby promote rapid reorganization of the actin filament network to drive epithelial wound closure.

- What are the noteworthy results?

The novel findings are that Swip-1 cross-links rather than bundles actin filaments in a Ca^{++} -dependent manner. Deletion of Swip-1 alters cell migration. The effects on migration differ in different cell types.

- Will the work be of significance to the field and related fields? How does it compare to the established literature? If the work is not original, please provide relevant references.

The work documents the function of Swip-1 both in vivo and in vitro. The wound healing assay supports a role for Swip-1 in lamellipodial protrusion accompanied by a transient increase in Ca^{++} .

We agree and thank the reviewer for these very positive constructive comments.

- Does the work support the conclusions and claims, or is additional evidence needed?

- Are there any flaws in the data analysis, interpretation and conclusions? - Do these

prohibit publication or require revision? The work supports the conclusions, but questions remain about whether the activity of Swip-1 functions as a brake of an accelerator of cell migration.

Our data confirmed a conserved role of Swip-1 in regulating lamellipodial protrusions and cell migration. As mentioned above, cell migration is a very plastic process and our data reflect the differential requirement of EFhD2/Swip-1 in distinct cell migration modes depending on distinct environments with different extracellular resistance.

The architecture of the actin cytoskeleton highly depends on substrate rigidity and cells adapt to different extracellular resistance by reorganizing their actin network (Lo et al., 2000; Mueller et al., 2017). B16-F1 cells migrate on a rigid 2D laminin substrate. Thus, without actin bundling and cross-linking activity of EFhD2, even in the presence of other cross-linkers, the Arp2/3 complex might be insufficient for maintaining the mechanical stability of the branched actin network at the leading edge of B16-F1 cells. Consequently, loss of EFhD2 in B16-F1 cells caused a significant reduction in protrusion rate, cell speed and effective migration. By contrast, immune cells migrate across a complex 3D environment with highly varying extracellular resistance that requires more dynamic protrusions. Thus, the loss of Swip-1 in *Drosophila* macrophages would result in a reduced actin network density that could lead to faster but less persistent migration *in vivo*. Supporting this notion, we found that forced overexpression of Swip-1 stabilizes protrusions at the expense of migration speed *in vivo*. Thus, the assembly of cross-linking proteins such as EFhD2/Swip-1 into the branched actin network might contribute to a resistance-adaptive behavior of migrating cells as recently proposed (Chen et al., 2020). In addition, EFhD2/Swip-1 has been recently identified as a cargo-specific adaptor for CG-endocytosis controlling integrin adhesion turnover (Moreno-Layseca et al., 2021a). This function might be also conserved and could further contribute to changed migratory behavior of macrophages *in vivo*. Interestingly, we now found that *swip-1* deficient macrophages showed an increase in integrin-marked adhesions. Enlarged focal adhesion caused by slower integrin turnover might further contribute to an increase in the migration speed as similarly observed for *zyxin*-deficient macrophages in *Drosophila* (Moreira et al., 2013). Such a prominent inverse relationship between focal adhesion size and cell migration speed has been shown in numerous motile cell types (Kim and Wirtz, 2013).

The authors tracked the movement of cells to determine the average speed of migration as a measure of Swip-1 role in cell migration. Unfortunately, the average speed of migration is not sufficient to capture the complexity of cell migration in different cell types and different environments. A more substantive analysis of the motility data including instantaneous velocity, pathlength and directionality might give clues to the difference observed between the 2 cell types.

We now provided a more substantial analysis of the motility data of macrophages including quantitation of displacement, straightness and track length (see new supplementary figure S2)

- Is the methodology sound? Does the work meet the expected standards in your field?

- Is there enough detail provided in the methods for the work to be reproduced?

The methodology is sound, and the details provided as sufficient to replicate the experiments.

We thank the reviewer for pointing this out.

Specific changes in the manuscript:

1. line 94 “shorter than implied”

The text has been changed accordingly.

2. line 127 “results explain”

The text has been changed accordingly.

3. line 184 Figure 2e, e’

The text has been changed accordingly.

4. line 186 Figure 2f

The text has been changed accordingly.

5. line 190 Figure 2g

The text has been changed accordingly.

6. line 321, the term leading edge is used incorrectly. Cells have leading edges, not the wound. Suggested wording "actin assembled to form broad lamellipodial protrusions in cells surrounding the wound site"

The text has been changed accordingly as follows "cells at the wound edge".

7. line 326 "edge of cells surrounding the wound"

The text has been changed accordingly.

8. line 330 ditto

The text has been changed accordingly.

9. line 332 "cells several rows back from the wound site exhibited a delayed response"

The text has been changed accordingly.

10. line 347, awkward phasing, clarify the direction of the change in $[Ca^{++}]$
References to figures 5 and 6 are misquoted in the text.

We apologize for this somewhat unfortunate wording and changed the text as follows: "However, immediately after wounding, a dramatic increase of intracellular calcium was observed as a bright fluorescence signal, first observed in front-row cells at the wound margin, but then rapidly spreading to more distal cell rows in the periphery within 20 seconds (Figure 4d; supplementary movie M11). After initial

calcium wave propagation, the levels of intracellular calcium first decreased in the periphery, then towards the wound margin within a minute (Supplementary movie M11).”

Reviewer #3 (Remarks to the Author):

The manuscript from Lehne et al describes a study centering on the sole *Drosophila* homolog of EFhD2/Swip-1 proteins, calcium-sensitive adapters that operate in diverse tissue and cell-type setting, but whose specific molecular functions, particularly vis-à-vis the actin-based cytoskeleton, have remained somewhat ambiguous. The study encompasses a rather broad range of topics and employs a variety of experimental approaches and techniques, which are carried out in competent fashion.

We thank the reviewer for pointing this out.

I wish to state at the outset of this review that, in my judgement, a primary drawback of the manuscript is a lack of focus, making it difficult for the reader to assess the novelty and significance of the reported findings. Two major, related advances are put forward (Figures 5-7 and related text): *Drosophila* EFhD2/SWIP-1 is shown to act as a microfilament cross-linker (rather than as a microfilament bundling protein), a capacity that is inhibited by calcium: EFhD2/SWIP-1 function is necessary for proper repair/closure of epithelial wounds in *Drosophila* pupae, a process involving calcium-regulated activity of the actin-based cytoskeleton. However, the first half or so of the manuscript is devoted to presentation of data which, to a large part, have been reported previously or are only marginally relevant to the key observations described and needlessly complicate matters (see below). I thus strongly recommend that the authors substantially re-organize and shorten the manuscript, as well as take into account specific additional issues detailed below.

We are grateful for this critical comment. For this reason, we have substantially restructured the manuscript. In particular, we have focused on presenting our data in

a more straightforward manner within the context of what has been previously published.

1. An aspect of EFhD2/SWIP-1 localization (via antibodies) that should be clarified (Figure 2 and accompanying text): Besides the general localization to the broad lamellipodia of pupal macrophages, EFhD2/SWIP-1 corresponds quite clearly - together with F-actin- to radial structures. I don't believe that these are mentioned in the text. Can the authors add comments on these structures? Are they perhaps associated with focal adhesions? Why are they observed only in the F-actin panel when visualized using super-resolution microscopy?

These F-actin rich radial bundles are frequently found in both primary macrophages isolated from pupae as well as in cultured macrophage-like S2R+ cells plated on concanavalin A (ConA)-coated surfaces. Endogenous Swip-1 localizes to these radial F-actin structures. Accumulation of endogenous Swip-1 to these radial structures often appears to be more striking on the confocal images (see new figure 1a'), whereas high-resolution SIM microscopy resolves the immunostaining much better and show a punctate localization of Swip-1 along these radial F-actin bundles (see merged SIM image in figure 1d'). However, these radial structures are not associated with focal adhesions. ConA-induced spreading is controlled by remodeling of the actin cytoskeleton upon binding of lectins to the polysaccharide side chains of plasma membrane proteins and lipids (Rogers et al., 2003), while cell spreading on uncoated glass surfaces or even better on vitronectin-coated surfaces depends on integrin-mediated cell adhesion (Jani and Schock, 2007; Nagel et al., 2017).

We thank the reviewer for this intriguing question. We therefore followed up a possible role of Swip-1 in regulating integrin adhesion as recently reported for a human breast cancer cell line (Moreno-Layseca et al., 2021). Indeed, we now provided additional evidence for a conserved function of EFHD2/Swip-1 regulating both lamellipodial actin networks and integrin adhesion dynamics. As described above, we found that isolated *swip-1* mutant macrophages exhibited increased β -integrin-marked focal adhesion sites resulting in an increased cell spreading (cell size), when plated on surfaces coated with the ECM protein vitronectin (see new

supplementary figure S2). Thus, enlarged focal adhesions caused by slower integrin turnover might further contribute to an increase in the migration speed as similarly observed for *zyxin*-deficient macrophages in *Drosophila* (Moreira et al., 2013). Such a prominent inverse relationship between focal adhesion size and cell migration speed has been shown in numerous motile cell types (Kim et al., 2013) and it might be also relevant in B-cells (Reimer et al., 2020 Cell Reports). We discuss this as an additional mechanism explaining an increase of cell migration speed of mutant immune cells *in vivo* (see discussion).

2. Figures 3 and 4 and the accompanying text describe experiments primarily designed to assess the functional role(s) of *Drosophila* EFhD2/SWIP-1 in pupal macrophages and of mouse EFhD2/SWIP-1 in a melanoma cell line. In keeping with the comments above, I found much of these data to be problematic on various levels. The reported knockout phenotypes, even if statistically significant, are not particularly compelling, do not provide true insight, and are in some cases perplexing and difficult to explain (eg- the opposite effects on motility in the fly and murine systems). Furthermore, previous publications (from the GIST (Gwangju) groups and others) have addressed much of what is reported here regarding mammalian EFhD2/SWIP-1, even if somewhat less rigorously.

We have restructured our manuscript. In particular, we now focus on presenting our data in a more straightforward manner within the context of what has been previously published. Notably, we considered previous publications in the introduction and in the results in the context of our data, in particular from the GIST (Gwangju) groups previously describing a role of EFHD2/Swip-1 in lamellipodia formation and cell migration of cultured B16-F1 mouse melanoma cells (Kwon et al., 2013, PlosOne; Huh et al., Cell Mol Life Sci 2013, Huh et al., Oncotarget 2015).

However, we do not share the opinion of the referee, that the reported knockout phenotypes, even if statistically significant, are not particularly compelling. Although single-cell analyses published to date clearly describe defects in lamellipodium formation and cell migration upon depletion of EFhD2 (Huh et al., Cell Mol Life Sci 2013, Huh et al., Oncotarget 2015), they still fall short of explaining the underlying molecular mechanism. The interpretation of these results is additionally clouded by incomplete description of experimental conditions in the Huh et al., 2015 study, as it

remains entirely unclear whether motility of EFhD2-depleted B16-F10 cells (see Fig 4 E and F in that study) was analyzed in Matrigel or in medium, and if the latter were the case, which ECM substrate was used. We feel that our analyses go substantially beyond this stage and provides a more rigorous analysis including CRISPR/Cas9 KO data of EFhD2.

According to our knowledge, we show for the first time that loss of EFhD2, dramatically diminishes lamellipodium width by about 40% in two independent clones (now shown in figure 6g). Since 2D migration on flat surfaces is primarily driven by actin assembly in the lamellipodium, this observation already explains the reduced cell migration speed. To substantiate these findings, we additionally recorded wild-type and EFhD2-deficient cells randomly migrating on laminin by time-lapse, phase-contrast microscopy, and determined respective protrusion rates by kymograph analyses (now shown in figure 6i, j). Quantification revealed cell edge protrusion to be reduced by about 30% in both mutants compared to the B16-F1 control (now shown figure 6k). Our data therefore strongly suggest that cross-linking of adjacent actin filaments within dendritic arrays by EFhD2 significantly contributes to mechanical stability of lamellipodia to drive efficient protrusion. Combined with the detailed biochemical analyses of EFhD2 *in vitro*, we feel that these findings are a major advancement in our molecular understanding of EFhD2 in the cellular context of fly and mammalian cells.

3. A major claim made here is that EFhD2/SWIP-1 acts to cross-link rather than bundle microfilaments, and does so in a calcium-dependent manner. The key experiment (Figure 5h and movie M4) involves visualization of microfilament network morphologies using TIRF microscopy, in which *Drosophila* EFhD2/SWIP-1 function is compared with two established (human) actin bundling proteins, fascin and alpha actinin 4 (this established characteristic of fascin and actinin 4 should be better emphasized in the text). Given that these novel findings contrast with previous studies of human EFhD2/SWIP-1, as acknowledged by the authors, the human homolog should be subjected to this assay as well.

This is an excellent idea as well as an import control. As suggested, we therefore purified recombinant human EFHD2 and compared its activity on actin filament architecture by TIRF microscopy. Consistent with previous observation, we detected prominent actin bundling activity. However, contrary to previous assumptions, this actin bundling activity was not regulated by calcium. Most importantly, we found that human EFHD2 protein exhibit a dual *in vitro* activity capable of bundling filaments into loose bundles and cross-linking of filaments into networks. Most importantly, we found that the cross-linking activity is calcium-sensitive. Thus, comparable to its fly counterpart Swip-1, quantitative analysis of cross-link events revealed that in the absence of calcium, Swip-1 and EFhD2 induced stable, perpendicular cross-links between elongating actin filaments, creating a stable and highly cross-linked network over time. On the other hand, EFhD2/Swip1 induced cross-links between actin filaments were rare or nearly absent in the presence of calcium (see Supplementary figure S3).

4. Returning now to the earlier remarks regarding manuscript organization. I believe that the manuscript will be vastly improved by revising and shortening the main text and associated data, so as to focus primarily on what now constitutes the second half of the paper (Figs 5-8). The material that currently appears earlier is, to my mind, inconclusive and distracting, and should be used sparingly, primarily to introduce EFhD2/SWIP-1 proteins and the *Drosophila* homolog. As noted, the experiments involving the mouse cell line are particularly problematic and, in my judgement, should not be included at all. Re-organizing the manuscript in this fashion is necessary to allow readers to recognize the novel observations reported and properly appreciate their significance.

As suggested, we restructured our manuscript, particularly we now focus on presenting our data in a more straightforward manner within the context of what has been previously published. However, we disagree with the reviewer's suggestions excluding our data on cultured B16-F1 mouse melanoma cells. Our data together provides strong evidence for a conserved function of EFhD2/Swip-1 between flies and mammals, but also highlights its differential function in regulating lamellipodial protrusions and cell migration depending on the cellular context. Mechanistically, our model (depicted in new figure 7) implies that EFhD2/Swip-1 changes the mechanical

properties of the actin cytoskeleton that allow cells to adapt in response to extracellular resistance (2D and 3D migration) but also in response to intracellular calcium changes (wound healing). Our data further indicates that both stabilizing and remodeling of WRC-Arp2/3-dependent branched actin filament networks require the actin cross-linking activity of EFhD2/Swip-1, conserved between human and flies. In the light of these new data, we still included our data on cultured B16-F1 mouse melanoma cells as main figure 6.

Minor issues

Figure 2e- what are the arrows pointing to? Also- the related text (line 184) erroneously refers to Figure 1 rather than Figure 2.

We apologize for this mistake and have changed the text accordingly. The arrows mark the lamellipodial localization of Swip-1-eEGFP in a protruding macrophage *in vivo*.

Figure 2f- Swip-1-eEGFP should be changed to Swip-1-eEGP.

We changed the text accordingly.

Lines 338-339- The authors reference two previous *Drosophila* studies that described calcium influxes in the context of wound healing, stating that both involved the embryonic epidermis. While this is true of Razzell et al, the Antunes et al study (reference #48) focused on the pupal notum.

Thank you. We have changed the text accordingly.

Figure 6d- what are the white asterisks marking?

White asterisks mark specialized epidermal cells, the so-called tendon cells providing attachment sides to the muscles. These tendon cells lack Swip-1-eGFP expression driven by the *da*-Gal4 driver. We now included this information in the legend of figure 4d.

References:

- Chen, X. *et al.* Predictive assembling model reveals the self-adaptive elastic properties of lamellipodial actin networks for cell migration. *Commun Biol* **3**, 616, doi:10.1038/s42003-020-01335-z (2020).
- Huh, Y. H. *et al.* Swiprosin-1 modulates actin dynamics by regulating the F-actin accessibility to cofilin. *Cell Mol Life Sci* **70**, 4841-4854, doi:10.1007/s00018-013-1447-5 (2013).
- Huh, Y. H. *et al.* Swiprosin-1 stimulates cancer invasion and metastasis by increasing the Rho family of GTPase signaling. *Oncotarget* **6**, 13060-13071, doi:10.18632/oncotarget.3637 (2015).
- Jani, K., Schock, F., 2007. Zasp is required for the assembly of functional integrin adhesion sites. *J Cell Biol* **179**, 1583-1597.
- Jumper, J. *et al.* Highly accurate protein structure prediction with AlphaFold. *Nature* **596**, 583-589, doi:10.1038/s41586-021-03819-2 (2021).
- Kim, D. H. & Wirtz, D. Predicting how cells spread and migrate: focal adhesion size does matter. *Cell Adh Migr* **7**, 293-296, doi:10.4161/cam.24804 (2013).
- Kwon, M. S. *et al.* Swiprosin-1 Is a Novel Actin Bundling Protein That Regulates Cell Spreading and Migration. *Plos One* **8**, doi:ARTN e716210.1371/journal.pone.0071626 (2013).
- Lo, C. M., Wang, H. B., Dembo, M. & Wang, Y. L. Cell movement is guided by the rigidity of the substrate. *Biophys J* **79**, 144-152, doi:10.1016/S0006-3495(00)76279-5 (2000).
- Moreira, C. G., Jacinto, A. & Prag, S. Drosophila integrin adhesion complexes are essential for hemocyte migration in vivo. *Biol Open* **2**, 795-801, doi:10.1242/bio.20134564 (2013).
- Moreno-Layseca, P. *et al.* Cargo-specific recruitment in clathrin- and dynamin-independent endocytosis. *Nat Cell Biol* **23**, 1073-1084, doi:10.1038/s41556-021-00767-x (2021).
- Mueller, J. *et al.* Load Adaptation of Lamellipodial Actin Networks. *Cell* **171**, 188-200 e116, doi:10.1016/j.cell.2017.07.051 (2017).
- Nagel B.M., Bechtold M., Rodriguez L.G. and **Bogdan S.** (2017) Drosophila WASH is required for integrin-mediated cell adhesion, cell motility and lysosomal neutralization. *J Cell Sci.* **130**: 344-359.
- Reimer, D. *et al.* B Cell Speed and B-FDC Contacts in Germinal Centers Determine Plasma Cell Output via Swiprosin-1/EFhd2. *Cell Rep* **32**, 108030, doi:10.1016/j.celrep.2020.108030 (2020).
- Rogers, S.L., Wiedemann, U., Stuurman, N., Vale, R.D., 2003. Molecular requirements for actin-based lamella formation in Drosophila S2 cells. *J Cell Biol* **162**, 1079-1088.

Reviewers' Comments:

Reviewer #1:

Remarks to the Author:

The authors have addressed all my concerns. The new data with the calcium-binding deficient mutant and the comparison with the human EFHD2 (as requested by another reviewer) are interesting and further increase the impact and novelty of the study.

Reviewer #2:

Remarks to the Author:

The concerns raised by this reviewer have been addressed appropriately. The manuscript is improved and the data shown in the figures support the conclusions. The substantive restructuring of the manuscript adds clarity and focus.

Reviewer #3:

Remarks to the Author:

1. In the revised manuscript, Lehne et al make a serious attempt at answering the concerns I raised regarding their initial submission. The manuscript text has been substantially modified and the text and figures reorganized, so as to better focus on the main points that the authors try to get across. The authors chose to keep the data on the mouse melanoma cell line, which I recommended be removed altogether, but separated it from the Drosophila data and moved it to the end of the Results section, a solution which I find acceptable. Some new data is now reported, mostly in response to points raised by the other reviewers. Somewhat enigmatically, the reorganization of the manuscript has resulted in a smaller number of main figures, but in a substantial increase in the overall length of the text. The authors may well want to re-examine the text and seek areas which can be shortened. In general, however, the manuscript reads much better now.

2. The authors responded to my initial review and corrected various minor problems, but in some cases have done so incompletely and these items should be better addressed:

- The authors explained in their response that the arrows in Figure 1e' "mark the lamellipodial localization of Swip-1-eEFP"- they should say so in the figure legend text.
- In Figure 1f, the labeling on the panels has remained "eEFP" rather than "eGFP".
- Line 360 should state "...embryonic and pupal wound closure", as was corrected in line 335.

REVIEWERS' COMMENTS

Reviewer #1 (Remarks to the Author):

The authors have addressed all my concerns. The new data with the calcium-binding deficient mutant and the comparison with the human EFHD2 (as requested by another reviewer) are interesting and further increase the impact and novelty of the study.

We thank the reviewer for pointing this out.

Reviewer #2 (Remarks to the Author):

The concerns raised by this reviewer have been addressed appropriately. The manuscript is improved and the data shown in the figures support the conclusions. The substantive restructuring of the manuscript adds clarity and focus.

We thank the reviewer for pointing this out.

Reviewer #3 (Remarks to the Author):

1. In the revised manuscript, Lehne et al make a serious attempt at answering the concerns I raised regarding their initial submission. The manuscript text has been substantially modified and the text and figures reorganized, so as to better focus on the main points that the authors try to get across. The authors chose to keep the data on the mouse melanoma cell line, which I recommended be removed altogether, but separated it from the Drosophila data and moved it to the end of the Results section, a solution which I find acceptable. Some new data is now reported, mostly in response to points raised by the other reviewers. Somewhat enigmatically, the reorganization of the manuscript has resulted in a smaller number of main figures, but in a substantial increase in the overall length of the text. The authors may well

want to re-examine the text and seek areas which can be shortened. In general, however, the manuscript reads much better now.

We thank the reviewer for pointing this out.

2. The authors responded to my initial review and corrected various minor problems, but in some cases have done so incompletely and these items should be better addressed:

- The authors explained in their response that the arrows in Figure 1e' "mark the lamellipodial localization of Swip-1-eEFP"- they should say so in the figure legend text.

We included the missing information in the figure legend (see line 1094f).

- In Figure 1f, the labeling on the panels has remained "eEFP" rather than "eGFP".

We changed this.

- Line 360 should state "...embryonic and pupal wound closure", as was corrected in line 335.

We changed this.